# Chasing All-Round Graph Representation Robustness: Model, Training, and Optimization

**Chunhui Zhang[1], Yijun Tian[2], Mingxuan Ju[2], Zheyuan Liu[1],**
**Yanfang Ye[2], Nitesh V. Chawla[2], Chuxu Zhang[1]***
[1]Brandeis University, {chunhuizhang,zheyuanliu,chuxuzhang}@brandeis.edu
[2]University of Notre Dame, {yijun.tian,mju2,yye7,nchawla}@nd.edu

## Abstract

Graph Neural Networks (GNNs) have achieved state-of-the-art results on a variety of graph learning tasks, however, it has been demonstrated that they are vulnerable to adversarial attacks, raising serious security concerns. A lot of studies have been developed to train GNNs in a noisy environment and increase their robustness against adversarial attacks. However, existing methods have not uncovered a principled difficulty: the convoluted mixture distribution between clean and attacked data samples, which leads to sub-optimal model design and limits their frameworks' robustness. In this work, we first begin by identifying the root cause of mixture distribution, then, for tackling it, we propose a novel method GAME - *Graph Adversarial Mixture of Experts* to enlarge model capacity and enrich the representation diversity of adversarial samples, from three perspectives of model, training, and optimization. Specifically, we *first* propose a plug-and-play GAME layer that can be easily incorporated into any GNNs and enhance their adversarial learning capabilities. *Second*, we design a decoupling-based graph adversarial training in which the component of the model used to generate adversarial graphs is separated from the component used to update weights. *Third*, we introduce a graph diversity regularization that enables the model to learn diverse representation and further improves model performance. Extensive experiments demonstrate the effectiveness and advantages of GAME over the state-of-the-art adversarial training methods across various datasets given different attacks.

## 1 Introduction

Graph neural networks (GNNs) have been demonstrated to be effective at learning from graphs. They explore a message-passing mechanism to update node representations by iteratively aggregating information from their neighbors, allowing GNNs to achieve state-of-the-art performance (Kipf & Welling, 2017; Veličković et al., 2018; Hamilton et al., 2017). Many real-world applications are based on GNNs, such as modeling over social networks (Fan et al., 2022; Zhang et al., 2019; Hu et al., 2020), scene graph reasoning (Chen et al., 2020; Zhang et al., 2022), and biological molecules (Jin et al., 2018; Xu et al., 2019; Guo et al., 2022).

Nevertheless, despite their outstanding performance, GNNs are susceptible to perturbations (Zügner et al., 2018b; Zügner & Günnemann, 2019; Zheng et al., 2021; Yue et al., 2022), which necessitate techniques to leverage GNN's robustness against adversarial attacks. Attackers can downgrade the performance of GNNs from multiple perspectives, such as adding or removing edges (Geisler et al., 2021; Chen et al., 2023), perturbing node properties (Zügner & Günnemann, 2019; Sun et al., 2020; Tian et al., 2023), and injecting malicious nodes (Zou et al., 2021; Ju et al., 2023). To enhance GNN's robustness, multiple defense methods against graph attacks have been proposed (Jin et al., 2020; Entezari et al., 2020; Zhang & Zitnik, 2020). However, most existing methods have not uncovered the principled difficulty (i.e., the convoluted mixture distribution between clean and attacked data samples), which results in sub-optimal model design, poor robustness, and limited performance. In light of this, we study the robustness of GNNs from a more fundamental perspective by discovering the key pattern behind the adversarial attacks that jeopardizes the performance of GNNs.

---

*Corresponding author

Figure 1: The distributions of node representations generated by two GNNs trained over clean and adversarial graphs. In (a), these two distributions are extremely similar. In (b) and (c), as the model gets deeper, a progressively larger differentiation between the two distributions is observed.

We begin by comparing the statistical differentiation between the latent representations of nodes on the clean graph and the adversarially generated graph, as shown in Figure 1. We observe that the distributions of node representations for clean and adversarial graphs before the message passing are highly similar (i.e., Figure 1(a)). However, as the model gets deeper, these two distributions get increasingly distinct, as demonstrated by the progressively larger shift shown from Figure 1 (a) to (c). This demonstrates that adversarial attacks imperil GNN's performance by generating adversarial graphs belonging to a distribution different from the clean graph, and the GNN model fails to transfer the knowledge learned from the clean graph to the generated adversarial graph.

To address the above challenge, we propose *Graph Adversarial Mixture of Experts* (GAME), a novel framework that enhances the robustness for GNN by expanding the model capacity and increasing the representation diversity for adversarial graphs. Specifically, we design GAME from three perspectives: ($i$) To strengthen the model capacity, we propose a plug-and-play GAME layer to accommodate the adversarial graphs with diverse mixture distributions by dynamically routing multiple assembled expert networks. ($ii$) From the training perspective, we present a decoupling graph adversarial training strategy, namely DECOG, where each expert network is trained by adversarial graphs generated by maximizing the gradient of other experts. DECOG enforces each expert to learn distinct distributions that all other experts under-perform at. ($iii$) From the optimization perspective, we incorporate a graph diversity regularization (GRADIV) to further enhance the diversity of knowledge learned via all expert networks such that GAME is capable of handling various adversarial graphs. GAME is an all-round robust framework that not only improves GNN's resilience to adversarial attacks, but also without too much extra cost compared with normal GNN, since GAME dynamically activates only one subset of the experts to participate in the computation. The contributions of this paper can be summarized as follows:

- To the best of our knowledge, this is the first work to improve GNN's robustness from the perspective of distribution differentiation. According to our empirical studies, existing GNNs fail to transfer the knowledge learned from one clean graph's distribution to another generated adversarial counterpart, which results in vulnerabilities to adversarial attacks.

- To solve this challenge, we propose an all-round framework, namely *Graph Adversarial Mixture of Experts* (GAME), from the perspectives of model design (i.e., GAME layer to bolster the model capacity), training (i.e., DECOG to diversify the adversarial graphs), and optimization (i.e., GRADIV to further diversify the experts' knowledge).

- Comprehensive experiments are performed on multiple benchmark datasets across varying scales, demonstrating that the robustness contributed to our proposed all-around GAME. The suggested method beats other common baselines in a variety of attack evaluations and natural evaluations, demonstrating that the all-around robust design of GAME handles intricate mixture distribution well and cleverly addresses a fundamental difficulty in graph adversarial training.

## 2 RELATED WORK

**Graph Neural Networks.** Graph Neural Networks have recently attracted a great deal of interest due to their effectiveness in learning non-Euclidean data and their remarkable performance in a vast array of graph mining tasks (Hamilton et al., 2017; Battaglia et al., 2018; Wu et al., 2020). Graph convolutional network (GCN) is proposed in the early stage of GNN research to apply the concept of convolution from image to graph data (Kipf & Welling, 2017; Gao et al., 2018; Wu et al., 2019a). Instead of simply averaging the features of neighboring nodes, graph attention networks (Veličković et al., 2018; Wang et al., 2019) use the attention module to value each neighboring node and learn

more important nodes during message passing. Simultaneously, skip connection is introduced to construct deeper GNNs and learn comprehensive representations in order to overcome the over-smoothing phenomenon (Li et al., 2019; 2021a;b). A concurrent work with ours exploits the sparsity property of the Mixture of Experts (MoE) mechanism to learn fairer representations on graphs (Liu et al., 2023). Unlike prior works focusing on improving model's standard accuracy or fairness, to the best of our knowledge, our GAME is the first attempt to improve the robustness of adversarial graphs by introducing the Mixture of Experts mechanism to strengthen GNN capacity.

**Adversarial Learning on Graphs.** It is demonstrated that deep learning models are susceptible to inputs with small adversarial perturbations, and several methods are proposed to improve the model's robustness (Goodfellow et al., 2015; Kurakin et al., 2017; Xie & Yuille, 2020). Recent research has shown that GNNs are susceptible to adversarial attacks without exception (Zheng et al., 2021), highlighting the urgent need to improve their robustness. Several methods are proposed for attacking graph data, including inserting or removing connections (Du et al., 2017; Chen et al., 2018; Waniek et al., 2018), perturbing node features (Zügner et al., 2018a; Zügner & Günnemann, 2019; Sun et al., 2020), or adding virtual nodes (Wang et al., 2020; Zou et al., 2021). In the meantime, numerous defense methods against graph attacks have been developed for learning robust GNNs (Zhu et al., 2019; Feng et al., 2020; Jin et al., 2020) or removing the attacked input during preprocessing (Entezari et al., 2020; Zhang & Zitnik, 2020). Go beyond prior works, this paper addresses a fundamental issue in graph adversarial learning (i.e., overly complex mixture distributions between clean and attacked nodes), subsequently improving the model capacity and representation diversity.

## 3 PRELIMINARIES

**Mixture of Experts.** In deep learning, the Mixture of Experts (MoE) constructs big neural networks with a dynamic routing strategy, which facilitates superior model capacity and attains better data parallelism (Shazeer et al., 2017). Given the input $x \in \mathbb{R}^d$, the current expert layer $\mathcal{E} = \{E_i(\cdot)\}_{i=1}^n$ with $n$ experts, and the gating network $P(\cdot) = \{p_i(\cdot)\}_{i=1}^n$, the output of MoE module can be formulated as follows:

$$y = \sum_{i \in \mathcal{T}} p_i(x) E_i(x), \tag{1}$$

where $\mathcal{T}$ represents the set of activated top-$k$ expert indices. In the above equation, the gating module $P(x)$ makes the activated portion of the model the same size as the normal network, hence enabling the efficient training of a larger neural network. Specifically, we calculate the gate-value $p_i(x)$ for $i$-th expert as follows:

$$p_i(x) = \frac{\exp(h(x)_i)}{\sum_{j=0}^N \exp(h(x)_j)}, \tag{2}$$

where $h(x)$ is a linear transformation to compute the logits of experts given input $x$, and $h(x)_i$ reflects the $i$-th value of the obtained logits, which weights the $i$-th expert in current layer.

**Adversarial Training.** Given the data distribution $\mathcal{D}$ over inputs $x \in \mathbb{R}^d$ and their labels $y \in \mathbb{R}^c$, a standard classifier (e.g., a neural network) $f : \mathbb{R}^d \to \mathbb{R}^c$ with parameter $\theta$ maps an input to labels for classification, utilizing empirical risk minimization $\min_\theta \mathbb{E}_{(x,y)\sim\mathcal{D}} \mathcal{L}(f(x;\theta),y)$, where $\mathcal{L}(\cdot,\cdot)$ represents the cross-entropy loss. Numerous strategies have been proposed to improve neural network adversarial robustness, with adversarial training-based methods being the most effective. The majority of cutting-edge adversarial training algorithms optimize a hybrid loss consisting of a standard classification loss and an adversarial loss term:

$$\mathcal{L}_{cls} = \mathcal{L}(f(x;\theta),y), \quad \mathcal{L}_{adv} = \max_{\delta \in \mathcal{B}(\epsilon)} \mathcal{L}(f(x+\delta;\theta),y), \tag{3}$$

where $\mathcal{L}_{cls}$ denotes the classification loss over standard (or clean) inputs, while $\mathcal{L}_{adv}$ is is the loss that encourages the model to learn from adversarial data samples, and $\mathcal{B}(\epsilon) = \{\delta \mid \|\delta\|_\infty \le \epsilon\}$ is the perturbation set. Popular adversarial training methods such as PGD (Madry et al., 2018) employs the same $\mathcal{L}_{cls}$ as in Equation 3, but substitute $\mathcal{L}_{adv}$ with a soft logits-pairing term. Thus, the overall goal of adversarial training is to minimize the following objective:

$$\min_\theta \mathbb{E}_{(x,y)\sim\mathcal{D}} [(1-\lambda)\mathcal{L}_{cls} + \lambda\mathcal{L}_{adv}], \tag{4}$$

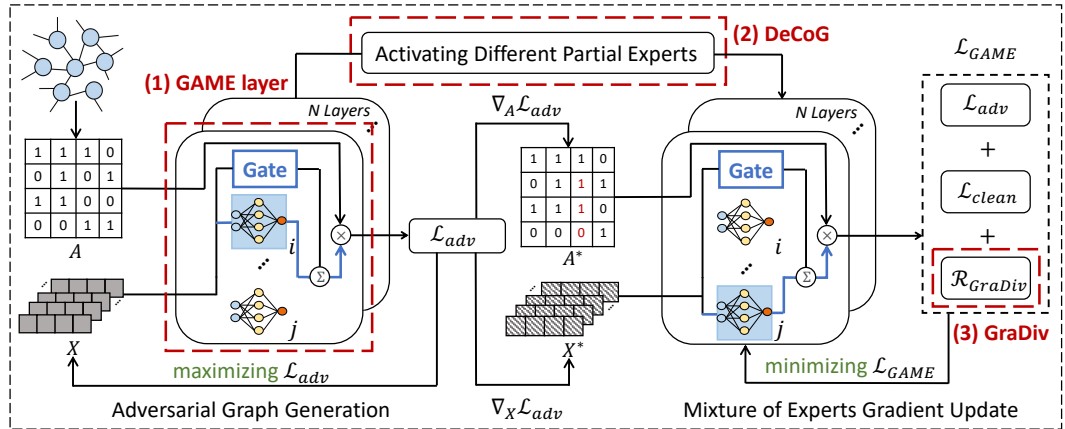

Figure 2: The illustration of the GAME framework: In (1) GAME layer, partial experts (in blue shaded region) are **activated** to compute adversarial gradient just for adversarial graph construction without weight updates (left part). Then, (2) DECOG decouples expert weights used for adversarial graph generation and model update, where the gate module **reactivates** other partial experts to fit created adversarial graphs in each GAME layer (right part). Finally, (3) GRADIV regularizes multiple experts to learn knowledge with more diversity. Note that in the left part, the model _maximizes_ adversarial loss to obtain the adversarial gradient only for graph perturbations. Next, in the right part, the model _minimizes_ overall loss on previously generated adversarial graph for weight updates.

where $\lambda$ is a fixed weight hyper-parameter. To balance standard and robust accuracies, the hyperparameter $\lambda$ must be set to a constant value during training for all of these contemporary adversarial learning works.

## 4 METHODOLOGY

In this section, motivated by the prior analysis of adversarial mixture distribution in Section 1, we present a novel framework GAME to increase the adversarial robustness of GNNs. Figure 2 illustrates the overall framework of the proposed model. Specifically, first, our model is developed using GAME layers, which increases model capacity on adversarial mixture distribution by introducing the MoE mechanism. Second, based on the GAME layer, we design a novel DECOG training strategy that augments more varied graphs to facilitate GNNs' adversarial training. Third, in order to classify adversarial samples from mixture distribution, we suggest graph diversity regularization for learning more distinguishable representations. The details of different parts are described in the following.

### 4.1 EXPANDING MODEL CAPACITY: GAME LAYER

To improve the learning capacity of GNNs and account for the overly complex mixture distribution, each GNN layer in our model includes a GAME layer at the model level. To update the feature of a target node, our model employs GAME layer to transform and aggregate the features of the neighboring nodes. Subsequently, the GAME combines the feature of target node and neighboring nodes to formulate node representation in the graph:

$$h_v^{(l)} = \text{COMB}^{(l)} \left( \text{GAME}^{(l)}(h_v^{(l-1)}), \text{AGGR} \left( \left\{ \text{GAME}^{(l)}(h_u^{(l-1)}), \forall u \in N_v \right\} \right) \right), \quad (5)$$

where $h_v^{(l)}$ denotes representation of node $v$ at $l$-th layer; $\text{AGGR}(\cdot)$ and $\text{COMB}(\cdot)$ represent the neighbor aggregation and combination functions, respectively. In Equation 5, the GAME layer are constructed by $\mathcal{W} = \{W_i(\cdot)\}_{i=1}^n$ and $P(\cdot)$, which represent the set of $n$ expert networks and the gate module, respectively. Then, the GAME layer is formulated as follows:

$$\text{GAME}(h) = \sum_{i \in \mathcal{T}} p_i(h) W_i(h), \quad (6)$$

where $\mathcal{T}$ represents the set of activated top-$k$ experts in each GAME layer. In comparison to conventional GNNs, the proposed sparse GAME layer is able to build a wider model with low computational cost, resulting in a greater capacity to express mixed distributions, as shown in Figure 2 (1).

## 4.2 Boosting adversarial graphs: DeCoG

For boosting the diversity of attacked node attributes and attacked adjacency matrices, we present Decoupling-based Graph Adversarial Training Strategy (DeCoG). DeCoG aims to deliver more robust performance than traditional graph adversarial training by transferring adversarial features from all experts to each individual expert. Specifically, given a clean graph $\mathcal{G}$ and the node labels $y$ with data distribution $\mathcal{D}$, we first calculate the loss function over both clean graph $\mathcal{G}$ and attacked graph $\mathcal{G}^*$ between input $\mathcal{G}$'s target nodes and their ground truth label $y$. Then, we learn a robust GAME model $f_{\boldsymbol{\theta}}$ with parameters $\boldsymbol{\theta}$:

$$\min_{\boldsymbol{\theta}} \mathbb{E}_{(\mathcal{G},y)\sim\mathcal{D}} (1-\lambda)\mathcal{L}_{ce}^{clean}(f_{\boldsymbol{\theta}}(\mathcal{G}),y) + \lambda\mathcal{L}_{ce}^{adv}(f_{\boldsymbol{\theta}}(\mathcal{G}^*),y), \tag{7}$$

where $\mathcal{L}_{ce}^{clean}(\cdot,\cdot)$, $\mathcal{L}_{ce}^{adv}(\cdot,\cdot)$ indicate the losses (e.g., cross-entropy for node classification) on clean graph and attacked graph, respectively, and $\lambda$ regulates the weight of the adversarial loss. The adversarial graph $\mathcal{G}^*$ is generated by GAME's customized PGD attack: during the learning procedure of PGD, GAME activates partial experts to compute multiple iterations of adversarial gradients. When the computation is finished, the gradients are added to the original graph as the final adversarial input for GNNs to minimize the adversarial learning loss. Therefore, each expert acquires the adversarial features generated by others. *From a high-level vantage point,* DeCoG *enables our* GAME *layer to implicitly transfer the aggregated knowledge of all experts to each individual expert.*

To formally describe the aforementioned pipeline, given the initial clean graph as $\mathcal{G}^{(0)}$, we first extract the initial adjacency matrix $A^{(0)}$ and node features $X^{(0)}$ from $\mathcal{G}^{(0)}$. Then, we dynamically sample the index set of activated experts $\mathcal{T}^0$ using $\{A^{(0)}, X^{(0)}\}$ to determine the GAME model's current activated part $f_{\boldsymbol{\theta}'^{(0)}|_{\mathcal{T}^0}}(\cdot)$, where $\boldsymbol{\theta}'^{(0)}|_{\mathcal{T}^0}$ is a subset of the GAME model $f_{\boldsymbol{\theta}}$ with parameters $\boldsymbol{\theta}$ at the current iteration. To maximize $\mathcal{L}_{ce}$, $f_{\boldsymbol{\theta}'^{(0)}|_{\mathcal{T}^0}}(\cdot)$ utilizes $\{A^{(0)}, X^{(0)}\}$ as input. The gradients computed for $A^{(0)}$ and $X^{(0)}$ are represented as $\nabla_{A^{(0)}}\mathcal{L}_{ce}(f_{\boldsymbol{\theta}'^{(0)}}(\mathcal{G}^{(0)}))$ and $\nabla_{X^{(0)}}\mathcal{L}_{ce}(f_{\boldsymbol{\theta}'^{(0)}}(\mathcal{G}^{(0)}),y)$, respectively. Both types of gradients are considered as adversarial noises, which are later incorporated to the current input $\{A^{(0)}, X^{(0)}\}$ to derive the perturbed adjacency matrix and node features $\{A^{(1)}, X^{(1)}\}$. The procedure can be formulated as:

$$A^{(t+1)} = \Pi_{\mathcal{B}(A,\epsilon)}(A^{(t)} + \alpha \cdot \text{sign}(\nabla_{A^{(t)}}\mathcal{L}_{ce}(f_{\boldsymbol{\theta}'^{(t)}}(\mathcal{G}^{(t)}),y))),$$
$$X^{(t+1)} = \Pi_{\mathcal{B}(X,\epsilon)}(X^{(t)} + \alpha \cdot \text{sign}(\nabla_{X^{(t)}}\mathcal{L}_{ce}(f_{\boldsymbol{\theta}'^{(t)}}(\mathcal{G}^{(t)}),y))), \tag{8}$$

where $\mathcal{B}(A,\epsilon)$ is the $\ell_0$ sphere with radius around clean $A$, $\mathcal{B}(X,\epsilon)$ is the $\ell_\infty$ sphere with radius around clean $X$, $\Pi_{\mathcal{B}(A,\epsilon)}$ means the nearest projection to $\mathcal{B}(A,\epsilon)$, $\Pi_{\mathcal{B}(X,\epsilon)}$ means the nearest projection to $\mathcal{B}(X,\epsilon)$, and $\alpha$ is the step size. With the aim of acquiring the adversarial graph $\mathcal{G}^* = \{A^{(*)}, X^{(*)}\}$, we calculate the gradients $T$ times as in Equation (8) and treat final gradients as perturbations. Later, $\mathcal{G}^*$ is utilized to train a different subset of activated experts. In general, DeCoG enables the dynamic activation of each expert in GAME and facilitates the computation of more diverse attacked graph adjacency matrices and node features, as depicted in Figure 2 (2).

## 4.3 Enhancing classifiability in robust representation: GraDiv

For empowering GAME in learning more distinguishable representation from the complex mixture distribution, we design Graph Diversity Regularization (GraDiv) from the optimization level. This regularization term penalizes the model to maximize the distance between node embeddings and enforces the model to learn distinct representations. The regularization is formulated as:

$$\mathcal{R}_{\text{GraDiv}} = -\log \sum_{i=1}^{N} \sum_{j=1}^{N} \frac{\exp(\text{Sim}(h_i, h_j)/\tau)}{\sum_{k=1}^{N} \exp(\text{Sim}(h_i, h_k)/\tau)}, k \neq i, \tag{9}$$

where $\text{Sim}(\cdot,\cdot)$ calculates the cosine similarity between two node representation, and $N$ denotes the number of nodes. $\mathcal{R}_{\text{GraDiv}}$ increases the distance between any pairs of learned node embeddings, which offers explicit supervision signal to learn node representations with more variety, and hence improving the model learning capacity. Finally, the overall learning objective function $\mathcal{L}_{\text{GAME}}$ is defined as a weighted combination of $\mathcal{L}_{ce}^{clean}$, $\mathcal{L}_{ce}^{adv}$, and $\mathcal{R}_{\text{GraDiv}}$:

$$\mathcal{L}_{\text{GAME}} = \mathcal{L}_{ce}^{clean} + \mathcal{L}_{ce}^{adv} + \mathcal{R}_{\text{GraDiv}}. \tag{10}$$

Consequently, $\mathcal{L}_{\text{GAME}}$ enables GAME to learn distinguishable node embeddings and preserve robustness to attacked node features, thereby improving the representation quality for downstream task.

## 5 EXPERIMENT

In this section, we perform comprehensive experiments on the graph robustness benchmarks to demonstrate the effectiveness of our proposed GAME model against adversarial graphs with complex distributions. This section is guided by answers to the following five research questions: **RQ-1**: Can GAME achieve better robustness compared to other SOTAs? **RQ-2**: How does each component in our all-around framework contribute to the robustness improvement? **RQ-3**: Can GAME separate the mixed distribution of clean and attacked nodes? **RQ-4**: Does GAME generate more diversified training adversarial graphs compared to baselines? and **RQ-5**: Can GAME learn distinguishable node representations?

### 5.1 EXPERIMENTAL SETUP

**Datasets**. We utilize Graph Robust Benchmark (Zheng et al., 2021) dataset to evaluate our model's performance by graphs with varying scales, including grb-cora (small-scale), grb-citeseer (small-scale), grb-flickr (medium-scale), grb-reddit (large-scale), and grb-aminer (large-scale).

**Baseline Methods**. We compare GAME with various baseline methods, spanning multiple perspectives. For models that specifically focus on robustness, we explore R-GCN (Zhu et al., 2019), GNN-SVD (Entezari et al., 2020), and GNNGuard (Zhang & Zitnik, 2020). In addition, we incoporate general GNN models (i.e., GCN (Kipf & Welling, 2017), GAT (Veličković et al., 2018), GIN (Xu et al., 2019), APPNP (Gasteiger et al., 2019a), TAGCN (Du et al., 2017), GraphSAGE (Hamilton et al., 2017), SGCN (Wu et al., 2019a)) with two generic defense approaches (i.e., layer normalization (Ba et al., 2016) (LN) and adversarial training (AT) (Madry et al., 2018)).

**Attacking Strategies**. We explore five effective yet diverse node injection attack methods to imperil the victim GNNs: RND (Zheng et al., 2021), FGSM (Goodfellow et al., 2015; Zheng et al., 2021), PGD (Madry et al., 2018), SPEIT (Zheng et al., 2021), TDGIA (Zou et al., 2021). These node injection methods have been proven to deliver *scalable* and *transferable* attacks (Zheng et al., 2021).

Details of attacking strategies and adversarial training are described in Appendix B.1 and B.2, respectively. Also, we include the statistics of datasets in Appendix C. In addition, edge modification attack evaluation based on Soft-Medoid/Soft-Median (Geisler et al., 2020; 2021) is in Appendix F.

### 5.2 OVERALL PERFORMANCE ON GRAPH ROBUST BENCHMARK

To answer **RQ-1**, we conduct the experiments to evaluate the robustness and report the performance in Figure 3 (We run 10 times for mean results/standard deviation and the train:val:test split is 0.6:0.1:0.3. Due to the page limitation, *we include the full results Table 2 with all numerical values in Appendix A*). As shown in Figure 7, GAME comprehensively achieves better robust accuracy than other baselines under five distinct attack assessments. For instance, on the small-scale grb-citeseer, the average accuracy of GAME outperforms the second-place R-GCN+AT by 2.96%; on the medium-scale grb-flickr, the average accuracy of our GAME outperforms the runner-up GAT+AT by 2.11%; on the large-scale grb-aminer, GAME outperforms the second-best GAT+AT by 1.32%. These results demonstrate the outstanding effectiveness of GAME against the distribution differentiation across five GRB graphs with different scales. Besides, under the *w.o. attack* setting, GAME significantly outperforms baselines by a large margin, proving that GAME equipped with multiple expert networks has stronger learning capability as well as higher accuracy on clean graphs. Though more parametrized than other baselines, GAME still enjoys the efficiency like regular GNNs and retains remarkable robustness against adversarial attacks, thanks to our proposed dynamic routing strategy that only activates partial experts to approximate the forward and backward of a normal GNN.

### 5.3 CONTRIBUTION OF INDIVIDUAL COMPONENT IN THE ALL-ROUND FRAMEWORK

GAME integrates three different components into a comprehensive graph robust learning framework, and to answer **RQ-2**, we conduct experiments on the performance without one of the individual components in GAME, denoted as $(a)$ w/o GAME layer, $(b)$ w/o DECOG, and $(c)$ w/o GRADIV, as shown in Table 1. We observe that the performance in both adversarial and clean graphs decreases after removing each component, demonstrating the contribution of each design in increasing the model's performance. For $(a)$ w/o GAME layer, removing GAME layer from our framework results

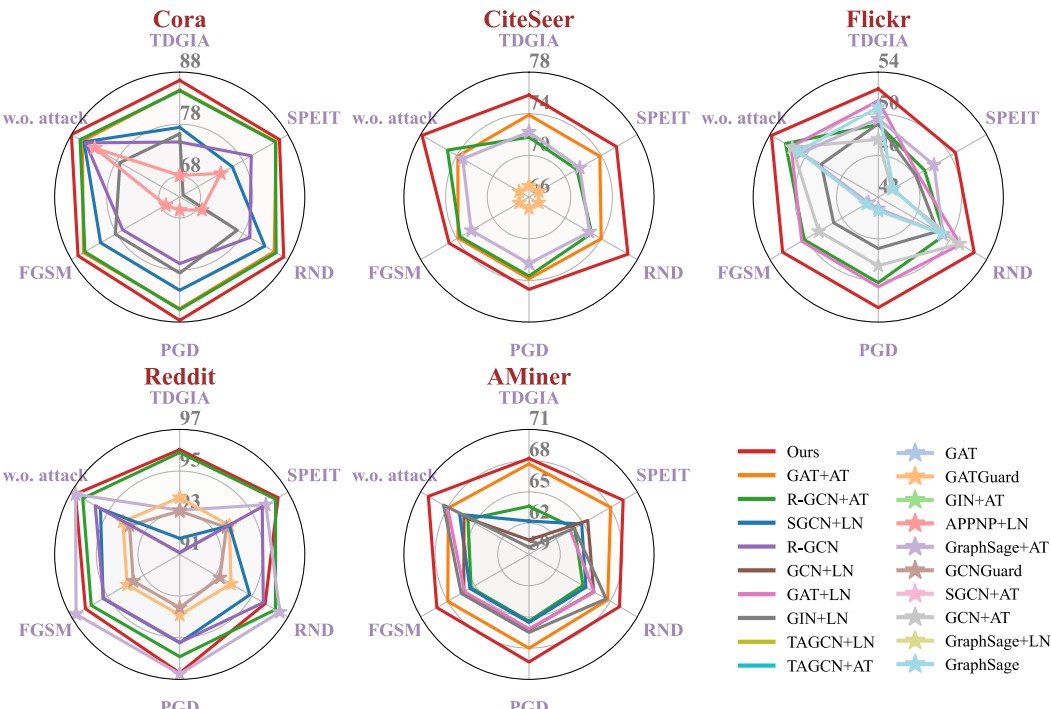

Figure 3: Overall assessments of all framework across graph with different scales. We apply TDGIA, SPEIT, RND, PGD-based, and FGSM-based graph attacks to evaluate the robustness of different frameworks. *w.o. attack* refers to the performance on clean graphs. For better clarity, we only include strong baselines in each figure.

in GAME being vanilla GCN on the model level. Such a removal disables the GRADIV training technique, as GRADIV depends on the GAME layer. Removing GAME layer decreases the performance on clean graphs (under w.o. attack) more than that on adversarial graphs. Specifically, under wo attack, this removal causes the model lose 2.58% accuracy, which is higher than the accuracy losses under two representative attacks (0.98% of PGD-based and 0.98% of FDSM-based) on grb-citeseer, respectively. For $(b)$ w/o DECoG, disabling the DECoG training strategy results in the generated adversarial graphs tightly coupled with the activated experts at the current iteration (i.e., the activated experts for adversarial gradient computation are identical to those updated by minimizing the loss). As a result, this removal causes a more severe performance deterioration for adversarial graphs than it does for clean graphs. Specifically, under PGD-based and FDSM-based attacks, this removal causes the model on grb-citeseer to lose 0.56% and 0.75% accuracy respectively, which is comparable to the removal of GAME layer. However, the accuracy loss on wo attack is only 1.56%, which is significantly less than the accuracy loss caused by GAME layer. It demonstrates that DECoG improves the performance of GNNs against adversarial attacks by generating diverse yet effective adversarial graphs, which is also compatible with the subsequent additional studies in Section 5.5. And for $(c)$ w/o GRADIV, GAME without the use of $\mathcal{R}_{\text{GRADIV}}$ in Equation 10 leads to a scenario where actively diversifying node embeddings is no longer an explicit supervision signal during the optimization process. As a result, this removal leads to performance downgrade on both clean graph and adversarial graph (0.65% on clean grb-citeseer and 0.42% on FGSM-attacked grb-citeseer, respectively), showing that the effectiveness of GRADIV in assisting GAME model to learn distinguishable representations on both clean and adversarial graphs.

## 5.4 Performance against Distribution Differentiation

We evaluate the performance of GAME and the vanilla GCN against the attacks from graphs generated through the adversarial distributions that are extremely divergent from the distribution of clean graphs, as shown in Figure 4. To answer **RQ-3**, we visualize the distributions of node representations from GCN and GAME at three stages (i.e., input layer (left column) and after first and second message passing layers (middle and right column respectively)). We observe that for the vanilla GCN, the distribution shift between the node representations in the clean and adversarial graphs still exists,

Table 1: Ablation studies for GAME on graphs with varying scales. Each row includes the model variant (mean result ± standard deviation) without one of the components in our all-around design.

| | Methods | TDGIA | SPEIT | RND | PGD-based | FGSM-based | w.o. attack | Avg. Acc. |
|---|---|---|---|---|---|---|---|---|
| *grb-citeseer* | GAME | **75.80** ± 0.99 | **75.69** ± 1.24 | **76.96** ± 1.04 | **74.86** ± 0.73 | **74.85** ± 0.71 | **77.86** ± 0.22 | **76.00** ± 0.82 |
| | w/o GAME layer | 73.96 ± 0.60 | 73.95 ± 0.34 | 74.21 ± 0.60 | 74.01 ± 0.52 | 73.89 ± 0.24 | 75.28 ± 0.09 | 74.22 ± 0.39 |
| | w/o DECOG | 74.07 ± 0.18 | 73.98 ± 0.83 | 74.70 ± 0.11 | 74.30 ± 0.82 | 74.10 ± 0.33 | 76.30 ± 0.80 | 74.58 ± 0.51 |
| | w/o GRADIV | 75.52 ± 0.28 | 75.50 ± 0.91 | 76.80 ± 0.51 | 74.42 ± 0.64 | 74.33 ± 0.46 | 77.21 ± 0.16 | 75.63 ± 0.49 |
| *grb-flickr* | GAME | **52.40** ± 1.02 | **50.54** ± 0.67 | **52.63** ± 0.87 | **52.62** ± 0.14 | **52.59** ± 1.20 | **53.83** ± 0.47 | **52.44** ± 0.73 |
| | w/o GAME layer | 51.32 ± 0.75 | 47.43 ± 0.14 | 51.02 ± 0.42 | 50.44 ± 0.55 | 50.60 ± 1.15 | 52.31 ± 0.22 | 50.52 ± 0.53 |
| | w/o DECOG | 51.63 ± 0.58 | 48.12 ± 0.45 | 51.52 ± 0.83 | 51.07 ± 0.83 | 49.95 ± 0.75 | 52.95 ± 0.89 | 50.87 ± 0.72 |
| | w/o GRADIV | 52.27 ± 0.18 | 50.14 ± 0.19 | 52.33 ± 0.28 | 52.19 ± 0.52 | 52.10 ± 0.85 | 53.14 ± 0.99 | 52.03 ± 0.50 |
| *grb-aminer* | GAME | 68.21 ± 1.27 | **69.43** ± 1.02 | **69.03** ± 0.72 | **69.33** ± 0.62 | **69.21** ± 0.69 | **70.15** ± 0.95 | **69.22** ± 0.88 |
| | w/o GAME layer | 67.89 ± 0.27 | 68.06 ± 0.30 | 67.95 ± 0.71 | 68.34 ± 0.29 | 68.20 ± 0.15 | 68.23 ± 0.40 | 68.11 ± 0.35 |
| | w/o DECOG | 67.96 ± 0.63 | 68.45 ± 0.48 | 68.22 ± 0.35 | 68.70 ± 0.18 | 68.25 ± 0.22 | 69.26 ± 0.48 | 68.47 ± 0.39 |
| | w/o GRADIV | 67.81 ± 0.78 | 69.32 ± 0.26 | 68.73 ± 0.39 | 69.12 ± 0.54 | 69.11 ± 0.41 | 69.66 ± 0.50 | 68.96 ± 0.48 |

Figure 4: The distributions of node representations on clean and adversarial graphs (*upper*: adversarial trained GCN; *lower*: GAME model) at the input layer (left), after the first batch normalization (BN) layer (middle), and after the last batch normalization (BN) layer (right). Each point indicates the mean and variance of a single channel in the BN layer.

even when the learning model forwards. In addition, the distribution difference between adversarial graphs and clean graphs is small. This phenomenon demonstrates that vanilla GNNs such as GCN suffer from mixture distributions between adversarial graphs and clean graphs, which prevents GCN from learning distinguishable representations. Compared with the vanilla GCN, GAME can successfully distinguish the node representations of the adversarial graphs from those of the clean graphs. Besides, as node representations pass through deeper layers, GAME still maintains the ability of discrimination, demonstrating GAME's outstanding effectiveness against adversarial graphs.

## 5.5 DIVERSITY OF ADVERSARIAL GRAPHS GENERATED BY GAME

To answer **RQ-4**, we visualize the distributions of both clean and adversarial graphs generated by GAME and GCN in Figure 5. We observe that the adversarial graphs generated from GCN are similarly distributed to the clean graph. On the contrary, GAME generates adversarial graphs whose distribution is statistically distinct from that of the clean graphs, further demonstrating the effectiveness

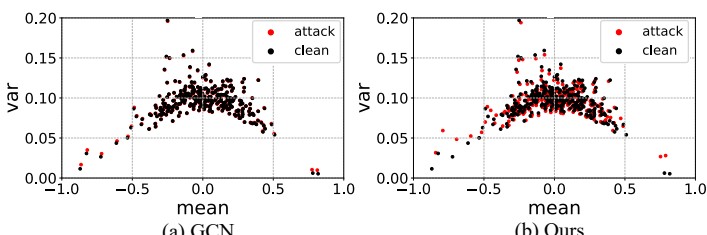

Figure 5: Distributions for clean and adversarial graphs created by GAME (b) and an adversarially trained GCN (a). For fair comparisons, we explore the same setting for both models.

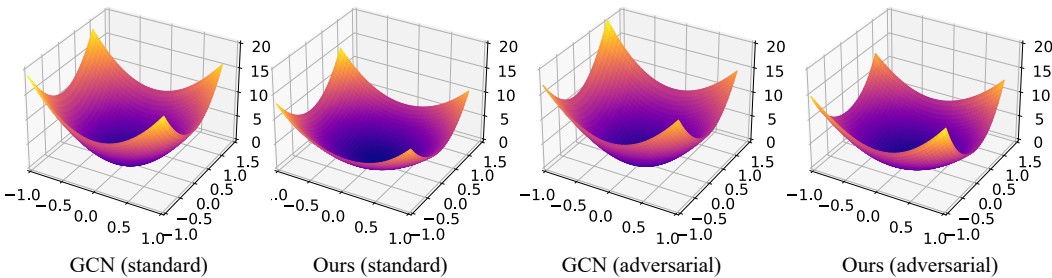

| GCN (standard) | Ours (standard) | GCN (adversarial) | Ours (adversarial) |

Figure 7: The loss landscapes of GAME and a vanilla GCN over clean graphs (the 1st and 2nd figures) and adversarial graphs (the 3rd and 4th figures). Under both settings, we visualize the same set of nodes randomly selected from the test set of the grb-cora dataset.

of DECoG and GRADIV. Diverse experts enable GAME to learn distinguishable node representations for robust performance, which significantly mitigates the GNN's training difficulties on the graphs with distinct distributions.

## 5.6 DIVERSITY OF NODE REPRESENTATIONS BY GAME

To answer **RQ-5**, on grb-cora dataset, we visualize the node representations generated by GAME and a vanilla GCN, shown in Figure 6. We observe that the node representations generated by GCN are generally entangled and intertwined with each other, while those generated by GAME exhibit a very well-clustered representation space with clear inter-cluster difference. This phenomenon demonstrates that GAME can distinguish the complex distributions and further learn distinguishable node representations, thanks to the all-around design of GAME.

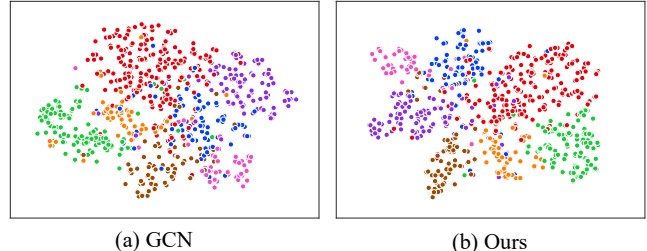

(a) GCN        (b) Ours

Figure 6: t-SNE (Van der Maaten & Hinton, 2008) for the representations of nodes in the test set in grb-cora. Nodes are colored according to their class labels.

## 5.7 ANALYSES ON OPTIMIZATION LANDSCAPE

To further validate the robustness of GAME, we analyze whether our all-round framework GAME reduces the difficulty of learning adversarial graphs by plotting its loss landscape (Li et al., 2018) w.r.t. both input and weight spaces. According to Figure 7, we observe that GAME leads to a flatter minima optimization landscape than the adversarially trained GCN on both clean graphs and adversarial graphs, indicating that the method advantageously alleviates the learning difficulty on the adversarial graph. our GAME reduces the complexity of learning adversarial graphs, allowing adversarial training model weights to be as simple as in a standard environment rather than GCN.

## 6 CONCLUSION

In this paper, we first identify the fundamental issue in adversarial graph learning: the mixture distribution between clean and attacked data samples. Motivated by this problem, we propose *Graph Adversarial Mixture of Experts* (**GAME**), a novel method to improve the model capacity, augment adversarial graphs, and enrich the graph representation diversity. For acquiring these triple improvements, GAME contains three innovative components, including a plug-and-play GAME layer, a decoupling graph adversarial training strategy DECoG, and a graph diversity regularization strategy GRADIV. GAME outperforms other baselines when evaluated on multiple datasets under different attack methods. Additional experimental analysis demonstrates the effectiveness of GAME in handling the complex mixture distribution, generating distinct adversarial graphs, and learning distinguishable representations.

## ETHICS STATEMENT

GAME enhances the robustness of GNN models against adversarial attacks, and therefore we believe that no ethical issues can be raised by our approach. In general, we should be very careful when applying machine learning models to ensure that there is no negative societal impact.

## REPRODUCIBILITY STATEMENT

To ensure the reproducibility of our experiments, we include the link of the source code in Appendix B.2. In addition, the hyper-parameters and other factors to reproduce our method are also provided in the Appendix B.2.

## ACKNOWLEDGMENTS

This work is partially supported by the NSF under grants IIS-2209814, IIS-2203262, IIS-2214376, IIS-2217239, OAC-2218762, CNS-2203261, CNS-2122631, CMMI-2146076, and the NIJ 2018-75-CX-0032. Any opinions, findings, and conclusions or recommendations expressed in this material are those of the authors and do not necessarily reflect the views of the funding agencies.

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

## A FULL RESULTS OF PERFORMANCE COMPARISON

We conduct extensive experiments on all five datasets in Table 2. Here we display the results of *graph injection* scenario with Top-5 attacks v.s. Top-10 defenses plus our GAME model. Since we have chosen strong defense methods as baselines, it is generally hard for attacks to be all effectives. The best performance is **bolded** and the runner-up is underlined.

Table 2: Main results on Graph Robust Benchmark datasets (i.e., *grb-cora*, *grb-citeseer*, *grb-flickr*, *grb-reddit* and *grb-aminer*). This table is a comprehensive supplementary to Figure 3. Partial results are cited from GRB (Zheng et al., 2021).

| | Methods | SPEIT | TDGIA | FGSM-based | PGD-based | RND | w.o. attack | Avg. Acc. |
|---|---|---|---|---|---|---|---|---|
| grb-cora | GAME | **86.10**±**0.81** | **86.42**±**0.61** | **86.53**±**0.72** | **87.68**±**0.62** | **87.04**±**0.58** | **88.00**±**0.73** | **86.96**±**0.68** |
| | R-GCN$_{+AT}$ | 85.21±0.41 | 84.43±0.27 | 85.60±0.38 | 85.01±0.41 | 85.36±0.41 | 86.07±0.00 | 85.28±0.15 |
| | GAT$_{+AT}$ | 85.35±0.19 | 84.55±0.50 | 85.43±0.34 | 85.33±0.72 | 84.95±0.58 | 85.57±0.00 | 85.20±0.21 |
| | SGCN$_{+LN}$ | 75.65±0.87 | 77.39±1.05 | 81.90±0.90 | 81.53±1.27 | 82.85±1.29 | 85.20±0.00 | 80.75±0.41 |
| | R-GCN | 79.85±0.48 | 74.58±1.76 | 76.77±0.74 | 76.62±0.90 | 79.53±0.74 | 84.83±0.00 | 78.69±0.27 |
| | TAGCN$_{+LN}$ | 71.73±1.14 | 79.67±1.53 | 83.22±0.60 | 83.00±0.61 | 84.22±0.58 | 85.07±0.00 | 81.15±0.42 |
| | GIN$_{+LN}$ | 64.75±0.62 | 76.14±1.80 | 78.58±0.51 | 78.26±0.88 | 76.75±0.89 | 77.24±0.00 | 75.29±0.49 |
| | APPNP$_{+LN}$ | 73.11±0.76 | 68.16±2.10 | 66.49±0.68 | 67.09±1.18 | 68.93±0.92 | 82.84±0.00 | 71.10±0.40 |
| | GIN$_{+AT}$ | 63.05±1.37 | 70.51±1.58 | 71.30±0.94 | 71.64±0.78 | 74.11±0.71 | 74.88±0.00 | 70.91±0.43 |
| | GATGuard | 65.67±0.00 | 65.67±0.00 | 65.67±0.00 | 65.67±0.00 | 65.67±0.00 | 65.67±0.00 | 65.67±0.00 |
| | GCN$_{+LN}$ | 59.45±0.64 | 72.58±2.71 | 78.21±0.68 | 77.60±1.14 | 81.34±0.60 | 83.58±0.00 | 75.46±0.49 |
| grb-citeseer | GAME | **75.69**±**1.24** | **75.80**±**0.99** | **74.85**±**0.71** | **74.86**±**0.73** | **76.96**±**1.04** | **77.86**±**0.22** | **76.00**±**0.82** |
| | R-GCN$_{+AT}$ | 71.30±0.64 | 71.75±0.78 | 73.63±0.50 | 73.58±0.36 | 72.95±0.68 | 75.03±0.00 | 73.04±0.17 |
| | GAT$_{+AT}$ | 73.85±0.22 | 73.89±0.40 | 73.86±0.28 | 73.84±0.26 | 74.00±0.45 | 73.88±0.00 | 73.89±0.11 |
| | SGCN$_{+LN}$ | 53.64±1.43 | 59.76±2.10 | 58.05±0.88 | 58.31±0.62 | 55.79±0.95 | 75.44±0.00 | 60.16±0.55 |
| | R-GCN | 59.32±3.65 | 56.69±1.77 | 55.07±1.43 | 54.90±1.58 | 60.43±0.71 | 73.16±0.09 | 59.93±0.57 |
| | TAGCN$_{+LN}$ | 51.09±6.38 | 46.87±4.99 | 63.39±1.21 | 64.25±1.48 | 73.06±0.58 | 73.56±0.00 | 62.04±1.70 |
| | GIN$_{+LN}$ | 52.82±0.93 | 52.16±2.47 | 61.36±1.03 | 61.60±0.95 | 65.66±0.43 | 66.04±0.00 | 59.94±0.63 |
| | SAGE$_{+AT}$ | 71.62±1.39 | 72.25±0.62 | 72.40±0.60 | 72.38±0.54 | 72.70±0.47 | 73.35±0.00 | 72.45±0.17 |
| | GIN$_{+AT}$ | 44.71±9.62 | 59.01±5.47 | 62.54±0.81 | 63.21±1.19 | 65.24±0.51 | 67.82±0.00 | 60.42±1.30 |
| | GATGuard | 67.08±0.00 | 67.08±0.00 | 67.08±0.00 | 67.08±0.00 | 67.08±0.00 | 67.08±0.00 | 67.08±0.00 |
| | GCNGuard | 64.54±0.13 | 64.58±0.00 | 64.56±0.04 | 64.46±0.16 | 64.55±0.16 | 64.58±0.00 | 64.54±0.04 |
| grb-flickr | GAME | **50.54**±**0.67** | 52.40±1.02 | 52.59±1.20 | 52.62±0.14 | 52.63±0.87 | 53.83±0.47 | **52.44**±**0.73** |
| | R-GCN$_{+AT}$ | 47.14±0.13 | 48.97±0.05 | 50.22±0.11 | 50.20±0.08 | 49.32±0.10 | 52.28±0.06 | 49.69±0.04 |
| | GAT$_{+LN}$ | 46.39±0.13 | 51.25±0.07 | 50.62±0.12 | 50.47±0.15 | 51.09±0.08 | 51.66±0.00 | 50.25±0.03 |
| | SAGE$_{+LN}$ | 47.62±0.04 | 47.13±0.07 | 47.56±0.07 | 47.58±0.08 | 47.47±0.07 | 47.47±0.00 | 47.47±0.03 |
| | SAGE$_{+AT}$ | 48.17±0.14 | 49.52±0.05 | 43.09±0.15 | 43.10±0.14 | 49.18±0.07 | 50.52±0.00 | 47.26±0.04 |
| | GCN$_{+AT}$ | 43.52±0.07 | 47.54±0.06 | 48.58±0.09 | 48.58±0.09 | 51.00±0.09 | 51.53±0.00 | 48.46±0.02 |
| | GIN$_{+LN}$ | 46.18±0.09 | 49.05±0.05 | 46.94±0.10 | 46.94±0.07 | 48.58±0.06 | 48.31±0.00 | 47.67±0.03 |
| | SAGE | 43.66±0.07 | 50.63±0.08 | 43.32±0.14 | 43.33±0.19 | 49.00±0.08 | 50.79±0.00 | 46.79±0.06 |
| | GIN$_{+AT}$ | 43.73±0.05 | 43.77±0.01 | 45.24±0.10 | 45.28±0.10 | 44.36±0.03 | 45.23±0.00 | 44.60±0.03 |
| | GAT | 49.91±0.17 | **52.46**±**0.09** | 42.90±0.24 | 42.78±0.27 | 50.84±0.20 | 50.12±0.00 | 48.17±0.09 |
| | APPNP$_{+LN}$ | 42.30±0.01 | 42.30±0.00 | 44.14±0.07 | 44.16±0.09 | 42.31±0.01 | 44.23±0.00 | 43.24±0.01 |
| grb-reddit | GAME | **96.45**±**0.20** | **96.03**±**0.45** | 96.22±0.28 | **96.69**±**0.38** | 95.73±0.37 | **96.78**±**0.32** | 96.31±0.01 |
| | GIN$_{+LN}$ | 96.31±0.02 | 95.92±0.01 | 96.48±0.02 | 96.48±0.01 | **96.60**±**0.01** | 96.68±0.00 | **96.41**±**0.33** |
| | TAGCN$_{+LN}$ | 96.31±0.02 | 95.89±0.01 | 96.91±0.01 | 95.90±0.01 | 96.30±0.02 | 96.37±0.00 | 96.11±0.01 |
| | TAGCN$_{+AT}$ | 95.77±0.02 | 93.12±0.01 | **96.74**±**0.01** | 96.74±0.01 | 96.54±0.01 | 96.77±0.00 | 95.14±0.01 |
| | GAT$_{+LN}$ | 93.76±0.02 | 93.73±0.01 | 93.91±0.02 | 93.91±0.01 | 93.84±0.02 | 94.15±0.00 | 93.88±0.01 |
| | R-GCN$_{+AT}$ | 93.59±0.01 | 93.08±0.01 | 93.56±0.01 | 93.57±0.01 | 93.23±0.02 | 93.78±0.00 | 93.47±0.01 |
| | TAGCN | 93.76±0.03 | 91.77±0.01 | 95.24±0.03 | 95.24±0.02 | 94.88±0.03 | 95.39±0.00 | 94.38±0.01 |
| | GCN$_{+LN}$ | 95.57±0.01 | 91.09±0.02 | 95.23±0.02 | 95.22±0.01 | 95.63±0.01 | 95.68±0.00 | 94.74±0.01 |
| | SAGE$_{+AT}$ | 90.21±0.02 | 90.16±0.01 | 90.54±0.02 | 90.54±0.02 | 90.37±0.03 | 90.48±0.00 | 90.38±0.01 |
| | SAGE | 92.96±0.04 | 85.98±0.03 | 93.75±0.02 | 93.78±0.03 | 94.59±0.02 | 95.16±0.00 | 92.70±0.01 |
| | SGCN$_{+LN}$ | 87.02±0.03 | 86.61±0.01 | 88.99±0.03 | 89.01±0.03 | 87.72±0.03 | 90.15±0.00 | 88.25±0.01 |
| grb-aminer | GAME | **69.43**±**1.02** | **68.21**±**1.27** | **69.21**±**0.69** | **69.33**±**0.62** | **69.03**±**0.72** | 70.15±0.95 | **69.22**±**0.88** |
| | GAT$_{+AT}$ | 68.04±0.03 | 67.69±0.03 | 68.01±0.02 | 68.00±0.02 | 67.72±0.04 | 67.93±0.00 | 67.90±0.01 |
| | R-GCN$_{+AT}$ | 64.05±0.04 | 63.62±0.32 | 65.41±0.01 | 65.41±0.02 | 64.98±0.02 | 65.76±0.00 | 64.87±0.05 |
| | SGCN$_{+LN}$ | 64.84±0.04 | 62.20±0.15 | 65.54±0.03 | 65.54±0.04 | 65.31±0.04 | 66.68±0.00 | 65.02±0.03 |
| | R-GCN | 64.06±0.04 | 61.99±0.22 | 65.05±0.04 | 65.05±0.04 | 64.45±0.04 | 65.85±0.00 | 64.41±0.04 |
| | GCN$_{+LN}$ | 65.51±0.02 | 60.38±1.46 | 66.22±0.02 | 66.22±0.02 | 66.17±0.02 | 66.20±0.00 | 65.12±0.25 |
| | GAT$_{+LN}$ | 64.02±0.04 | 59.69±1.57 | 66.49±0.04 | 66.50±0.06 | 67.54±0.04 | 68.47±0.00 | 65.45±0.26 |
| | GIN$_{+LN}$ | 63.11±0.02 | 59.59±0.42 | 64.63±0.04 | 64.65±0.04 | 64.36±0.06 | 65.59±0.00 | 63.65±0.07 |
| | TAGCN$_{+LN}$ | 62.59±0.04 | 59.06±1.75 | 64.82±0.04 | 64.82±0.03 | 64.33±0.03 | 64.91±0.00 | 63.42±0.29 |
| | TAGCN$_{+AT}$ | 63.77±0.06 | 57.24±5.04 | 66.32±0.02 | 66.34±0.03 | 66.42±0.03 | 67.08±0.00 | 64.53±0.84 |
| | GAT$_{+LN}$ | 63.58±0.06 | 56.63±6.75 | 66.14±0.04 | 66.15±0.06 | 66.23±0.04 | 68.02±0.00 | 64.46±1.13 |

# B  IMPLEMENTATION DETAILS

## B.1  REPRODUCIBILITY SETTINGS OF ATTACK METHODS

We evaluate all the methods using five most common graph attack methods provide by GRB benchmark (Zheng et al., 2021), including random, fast gradient sign method, projected gradient descent, SPEIT, and topological defective graph injection attack. The details of these five attack methods are as follows:

- **RND** (Random): a method that injects random noises generated by Gaussian distribution (Zügner et al., 2018a).

- **FGSM** (Fast Gradient Sign Method): a method that linearizes the loss function around the current value of parameters to get an optimal max-norm constrained perturbation (Goodfellow et al., 2015).

- **PGD** (Projected Gradient Descent): a first-order adversary method that generates strongest assault using local first-order information about the network (Madry et al., 2018).

- **SPEIT**: a winning solution of KDD-CUP 2020 Graph Adversarial Attack & Defense competition, which is a global black-box graph injection attack with adversarial adjacent matrix and feature gradient attacks (Qinkai et al., 2020).

- **TDGIA** (Topological Defective Graph Injection Attack): a powerful graph injection attack that injects malicious nodes progressively around topologically vulnerable nodes in the graph (Zou et al., 2021).

Following the prior work (Zheng et al., 2021), using a vanilla GCN as the surrogate model brings more transferable and better black-box node injection attack effects than other GNN models. Therefore, we choose GCN as the surrogate model for all attacks in our experiments.

## B.2  HYPER PARAMETERS AND ADVERSARIAL TRAINING DETAILS

The hyper-parameters of GAME are shown in Table 3. The Hyper-parameters for adversarial training used in DECOG are included in Table 4. We then show the the adversarial training (AT) procedure of GAME as follows:

Step ① Initialization: The warm-up step. The training graph is utilized to train GAME model for a few iterations.

Step ② PGD attack: The PGD attack is employed to inject the malicious nodes and edges that assault the training nodes by message passing and create an attacked graph.

Step ③ Update gradients: The model parameters are updated based on the gradients that are calculated by training on the attacked graph and minimizing the node classification loss.

Step ④ Repetition: This adversarial training procedure is repeated until the training loss converges. Finally, we can obtain a GAME model with better learning capability and outstanding robustness. According to Table 2, we intriguingly discover that GAME with PGD can also defend against other types of attacks, demonstrating the superior generality and applicability of GAME.

Notably, we discover that the GAME layer's favorable attribute, i.e., sparsely activating partial experts, can be employed to design DECOG training strategy, which augments more diverse graphs. Specifically, in the Step ② and Step ③, we activate *different* portions of experts in GAME model, despite the fact that both steps are performed in the same epoch. The code is provided this anonymous link [1]. In addition, the suggested GAME is scalable to graphs of varying sizes. We use mini-batch training via neighborhood sampling with batchsize 1k to train GAME on *grb-reddit* and *grb-aminer* datasets. In the future, we will transfer the framework of GAME to other GNN models.

Table 3: Hyper-parameters of GAME for *grb-cora*, *grb-citeseer*, *grb-flickr*, *grb-reddit* and *grb-aminer* datasets. The $n$ and $k$ represent the number of total experts and activated experts in each layer, respectively. Note that during generating adversarial samples, we activate all experts. Noisy rate controls the randomness when the gate module activates the partial experts during minimizing the loss.

| Model | Datasets | $n$ | $k$ | Hidden sizes | LR | Dropout | Optimizer | Noisy rate |
|---|---|---|---|---|---|---|---|---|
| | *grb-cora* | 2 | 1 | 64, 64, 64 | 0.01 | 0.5 | Adam | 1e-2 |
| | *grb-citeseer* | 2 | 1 | 64, 64, 64 | 0.01 | 0.5 | Adam | 1e-1 |
| GAME | *grb-flickr* | 3 | 1 | 128, 128, 128 | 0.01 | 0.5 | Adam | 1e-1 |
| | *grb-reddit* | 3 | 1 | 128, 128, 128 | 0.01 | 0.5 | Adam | 1e-2 |
| | *grb-aminer* | 3 | 2 | 64, 64, 64 | 0.01 | 0.5 | Adam | 1e-1 |

Table 4: Hyper-parameters of adversarial training in DECOG for *grb-cora*, *grb-citeseer*, *grb-flickr*, *grb-reddit* and *grb-aminer* datasets. Noisy rate controls the randomness when the gate module maximizes the loss and activates the partial experts. Nodes represents the number of injected nodes, and Edges indicates the number of added edges.

| Model | Datasets | Attack | Step size | Iter. | Nodes | Edges | Feature range | Noisy rate |
|---|---|---|---|---|---|---|---|---|
| | *grb-cora* | PGD | 0.01 | 10 | 20 | 20 | [-0.94, 0.94] | 1e-1 |
| | *grb-citeseer* | PGD | 0.01 | 10 | 30 | 20 | [-0.96, 0.89] | 1e-1 |
| GAME | *grb-flickr* | PGD | 0.01 | 10 | 200 | 100 | [-0.47, 0.99] | 2e-1 |
| | *grb-reddit* | PGD | 0.01 | 10 | 500 | 200 | [-0.98, 0.99] | 1e-1 |
| | *grb-aminer* | PGD | 0.01 | 10 | 500 | 100 | [-0.93, 0.93] | 2e-1 |

Table 5: Statistics of five GRB datasets covering from small- to large-scale graphs.

| Dataset | Scale | #Nodes | #Edges | #Feat. | #Classes | Feat. Range (original) | Feat. Range (normalized) |
|---|---|---|---|---|---|---|---|
| *grb-cora* | Small | 2,680 | 5,148 | 302 | 7 | [-2.30, 2.40] | [-0.94, 0.94] |
| *grb-citeseer* | Small | 3,191 | 4,172 | 768 | 6 | [-4.55, 1.67] | [-0.96, 0.89] |
| *grb-flickr* | Medium | 89,250 | 449,878 | 500 | 7 | [-0.90, 269.96] | [-0.47, 1.00] |
| *grb-reddit* | Large | 232,965 | 11,606,919 | 602 | 41 | [-28.19, 120.96] | [-0.98, 0.99] |
| *grb-aminer* | Large | 659,574 | 2,878,577 | 100 | 18 | [-1.74, 1.62] | [-0.93, 0.93] |

## C  STATISTICS OF GRAPH ROBUST BENCHMARK DATASETS

We evaluate our proposed GAME as well as adversarial learning framework on five real-world GRB datasets (Zheng et al., 2021), spanning from small- to large-scales. The data statistics are displayed in Table 5. To utilize *grb-cora, grb-citeseer, grb-flickr, grb-reddit, grb-aminer*, we apply the tool provided by Graph Robustness Benchmark [2].

## D  GRADIV'S EFFECTIVENESS ON DIVERSITY OF NODE REPRESENTATIONS

To discuss GRADIV's impact on nodes' representation diversity, on grb-cora dataset, we visualize the node representations generated by GAME without GRADIV and GAME, shown in Figure 8. We observe that the node representations generated by GAME without GRADIV are more entangled and intertwined with each other then GAME. This phenomenon demonstrates that GRADIV is able to learn more distinguishable node representations.

## E  ADVERSARIAL GRAPHS GENERATED BY GAME ON GRB-FLICKR

We observe that the adversarial graphs generated from GCN are similarly distributed to the clean graph in Figure 9 (i.e.,the red and black dots in the lower left corner of the GCN figure overlap significantly, but in the GAME figure the overlap is minor). On the contrary, GAME generates adversarial

---

[1] https://tinyurl.com/game23code
[2] https://github.com/thudm/grb

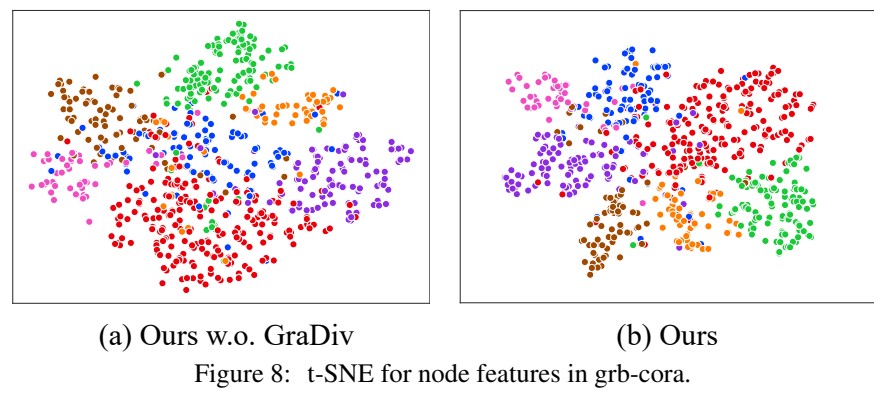

(a) Ours w.o. GraDiv                (b) Ours

Figure 8: t-SNE for node features in grb-cora.

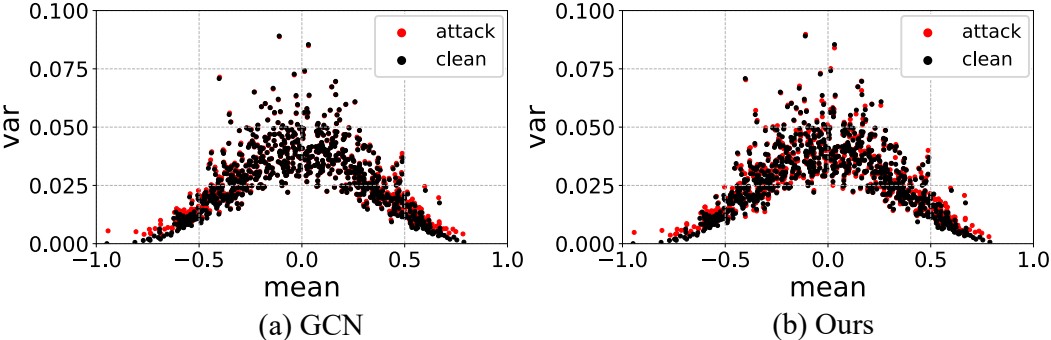

(a) GCN                (b) Ours

Figure 9: Distributions for clean and adversarial graphs created by GAME (b) and an adversarially trained GCN (a). For fair comparisons, we explore the same setting for both models on *grb-flickr*.

graphs whose distribution is statistically distinct from that of the clean graphs, further demonstrating the effectiveness of DECOG and GRADIV. Diverse experts enable GAME to learn distinguishable node representations for robust performance, which significantly mitigates the GNN's training difficulties on the graphs with distinct distributions.

## F  EVALUATION ON EDGE MODIFICATION ATTACK

We apply the proposed GAME framework into Soft-Medoid/Soft-Median GDCs (Geisler et al., 2020; 2021). Note that the edge Modification attack (Geisler et al., 2020; 2021) are also crucial for assessing GNN's robustness. Therefore, we present experimental results in Table 6, which are based on three random splits for rigorous comparisons. We observe that GAME enhances their accuracy.

Table 6: The robust accuracy of Soft-Median GDC and Soft-Medoid GDC without or with our GAME framework on the Cora dataset with the global attacks (GR-BCD & PR-BCD, $\epsilon = 0.1$) proposed by Soft-Medoid GDC (Geisler et al., 2021) . We set the number of experts to 10 and the hidden units of each expert to 32. We run them on three random splits and report the mean and standard error results.

|  | GR-BCD (Geisler et al., 2021) | PR-BCD (Geisler et al., 2021) |
|---|---|---|
| GCN (Kipf & Welling, 2017) | 0.622 ± 0.003 | 0.645 ± 0.002 |
| GDC (Gasteiger et al., 2019b) | 0.677 ± 0.005 | 0.674 ± 0.004 |
| PPRGo (Bojchevski et al., 2020) | 0.726 ± 0.002 | 0.700 ± 0.002 |
| SVD GCN (Entezari et al., 2020) | 0.755 ± 0.006 | 0.724 ± 0.006 |
| Jaccard GCN (Wu et al., 2019b) | 0.664 ± 0.001 | 0.667 ± 0.003 |
| RGCN (Zhu et al., 2019) | 0.665 ± 0.005 | 0.664 ± 0.004 |
| Soft-Median GDC (Geisler et al., 2020) | 0.765 ± 0.001 | 0.752 ± 0.002 |
| Soft-Medoid GDC (Geisler et al., 2021) | 0.775 ± 0.003 | 0.761 ± 0.003 |
| Soft-Median GDC (+GAME) | 0.772 ± 0.005 | 0.759 ± 0.005 |
| Soft-Medoid GDC (+GAME) | 0.780 ± 0.007 | 0.772 ± 0.006 |

