# OpenReview forum: "Chasing All-Round Graph Representation Robustness: Model, Training, and Optimization"
_ICLR.cc/2023/Conference — ICLR 2023 poster_

### Official Review · Reviewer_jUru · 2022-10-24

**Confidence:** 4
**Correctness:** 4
**Technical Novelty And Significance:** 3
**Empirical Novelty And Significance:** 3
**Recommendation:** 8

**Clarity, Quality, Novelty And Reproducibility:**

The motivation is clear. The structure and presentation of GAME in this paper are easy to understand. The proposed method is relatively new and novel. The authors also provide code and data in the appendix for reproducibility.

**Strength And Weaknesses:**

Strength:
1. This paper initially discovers an interesting distribution difference between common graph learning and adversarial graph learning. This difference also is one of the main reasons which makes adversarial graph learning become difficult. This discovery is interesting.

2. The proposed model couples adversarial training with the Mixture of Experts mechanisms. This joint design offers stronger adversarial attack samples for activated experts since the current adversarial noises are generated from other different inactivated experts. The model is relatively new and novel to me.

3. The authors provide comprehensive thoughts for stronger graph robustness. The investigation of model, optimization and training contributes to a thorough study. Also, the paper provides a deep and meaningful thought on the correlation between MoE and PGD.

4. Comprehensive evaluation across multiple scales of graph robust benchmark datasets are conducted. GAME outperforms many baseline methods in terms of robustness and accuracy. The authors also provide an in-depth analysis of GAME's capacity to mitigate the distribution difference between standard graph learning and adversarial graph learning. The presentation of this paper is overall good.

Weakness:
1. I tend to think that GAME has the potential for more general use, such as enhancing adversarial training on general machine learning. Since this paper is specifically focusing on graph adversarial training, authors need to clarify why GAME is designed for graph tasks (e.g., the exact benefits of GAME on graph tasks).

2. The authors provide empirical explanations about why the computational cost of the proposed GAME is close to the common GNN while this paper has no mathematical estimations about GAME's computation and the baseline GNN. Authors should provide better complexity analysis for a more convincing claim on GAME's efficiency.

3. The connection between the $R_{Gradiv}$ and backbone design is not very clear. Although experiments say that the gradiv regularization improves the model's performance, this regularization operation causes extra human-craft efforts in adjusting the hyperparameter.


**Summary Of The Paper:**

This paper aims to defend against evasion attacks in node-level graph learning. The authors introduce a Graph Adversarial Mixture of Experts (GAME) framework for learning robust representation in an adversarial training manner (e.g., PGD). In particular, GAME combines PGD training and the Mixture of Experts mechanism for ensembling adversarial gradients from different sub-models to strengthen the attack on the graph. The experiments further show that this unified design of GAME can be implemented as efficiently as normal GNN adversarial training but offers impressive advantages over prior work while under different graph attack evaluations.

**Summary Of The Review:**

This paper proposes to study graph adversarial learning. The proposed GAME method is novel and well-motivated. The extensive experiments practically support the claims in the paper and answer the questions raised with motivation. In summary, this is good and solid work. I tend to accept this paper.

---

> ### Author Response · Authors · 2022-11-13
> **Response to Reviewer jUru**
>
> Thank you for your constructive feedback. We reply to your comments as follows.
>
> **Q1**: I tend to think that GAME has the potential for more general use. Since this paper is specifically focusing on graph adversarial training, authors need to clarify why GAME is designed for graph tasks.
>
> **A1**: Thank you for acknowledging the potential of our approach in a variety of other different areas. Different from the adversarial training in other fields (e.g., CV, NLP), our GAME uses not only node features as the object of adversarial training, but also the adjacency matrix with extra structure information. It considers a more complex type of data for adversarial training (due to the geometric information of the graph).
>
>
> **Q2**: The authors provide empirical explanations about why the computational cost of the proposed GAME is close to the common GNN while this paper has no mathematical estimations about GAME's computation and the baseline GNN. Authors should provide better complexity analysis for a more convincing claim on GAME's efficiency.
>
> **A2**: Mathematical complexity analysis: we set the dimension of the hidden layer as $L$ for GNN. For our GAME model, we set $N$ as the total number of experts and $k$ as the number of activated experts, then we get $L=N\times k$. The computational complexity of normal GNN is $O(L)$ and the computational complexity of normal GNN is $O(k)$. If we fix the $k$ constant, the computation will not grow crazily like normal GNN while the dimension of the hidden layer grows.
>
> Empirical complexity analysis: computation measurements are shown in Table 1. According to Table 1, when the dimension of the model grows, the training time of our model increases slower than GCN since it always only activates $k$ experts during forward pass no matter the total width dimensions.
>
> Table 1: Training time of GCN and GAME with different dimensions. We set the width of a single expert as 32 and $32\times k$ is equal to the total dimension of our GAME model.
> | model   | GCN (32) | GCN (512) | GCN (1024) | GCN (4096) | GAME (32) | GAME (32*16=512) | GAME (32*32=1024) | GAME (32*128=4096) |
> |:-------:|:----------:|:---------:|:---------:|-------:|:----------:|:---------:|:---------:|:---------:|
> | training time | 05:19 | 05:40 | 08:59 | 41:54 | 05:38 | 05:40 | 05:59 | 10:19 |
>
>
> **Q3**: The connection between the gradiv and backbone design is not very clear. Although experiments say that the gradiv regularization improves the model's performance, this regularization operation causes extra human-craft efforts in adjusting the hyperparameter.
>
> **A3**: We don't need extra hyperparameter tuning on the GraDiv regularization since it can stably improve GAME's performance as following Table 2 shown. It shows that our method has a robust performance and is not sensitive to hyperparameters.
>
> Table 2: Results of GAME with different coefficients of GraDiv regularization on grb-cora dataset under FGSM attack.
> | Coefficient | 0.5 | 1.0 | 2.0 |
> |:-------:|:----------:|:---------:|:---------:|
> | Accuracy (\%) | 74.53 | 74.85 | 74.68 |

---

### Official Review · Reviewer_SpxS · 2022-10-24

**Confidence:** 5
**Correctness:** 3
**Technical Novelty And Significance:** 3
**Empirical Novelty And Significance:** 3
**Recommendation:** 5

**Clarity, Quality, Novelty And Reproducibility:**

**Clarity**: Mostly good, the presentation is clear and easy to follow.

**Quality**: I have some doubts about the significance of the contributions:
* GAME (expert) layer: according to the supplementary material, the number of experts is only 2-3, depending on the dataset -- and the number of sampled experts in the forward pass is often one. If the mixture of experts is indeed useful, I would expect a performance gain when using more experts. When only one expert is sampled in the forward pass, it could happen that only one of the two experts gets sampled all the time, reverting the model back to a vanilla one.
* GraDiv: the additional loss term has complexity $O(N^2)$. This is clearly infeasible for larger datasets, e.g., on the `grb-aminer` dataset this requires roughly 1.6TB of memory when performing full-batch training. There is no mentioning of batched training, so it is unclear how this additional loss term is used.
* DeCoG: What is the difference to standard PGD attacks [Xu et al. 2018]? The fact that only a subset of the GAME model is activated in the forward pass comes from the fact the GAME layer is used, not from the DeCoG contribution itself.
* Using infinity-norm is meaningless on graph adjacency matrices because they are discrete, i.e., either we allow no edge to change (epsilon=0) or all edges can change at the same time (epsilon=1).

Regarding the experiments:
* Figures 6 and 7 are purely qualitative and do not strongly show an advantage of the proposed method over GCN.
* Figure 4 is confusing. In the top row (GCN) the adversarial and clean representations are much more similar than in the bottom row, contrary to the description in the text. This figure shows an advantage of GCN over the proposed method.
* Similarly in Figure 5, both plots look very similar and it is not clear why the proposed method has an advantage over GCN.
* Table 2: What do the confidence intervals correspond to? Is it standard error/ standard deviation? Over how many training runs? How many different random splits? What is the size of the splits?
* Why are the confidence intervals for "w.o. attack" almost always exactly zero for the baselines?
* Why are the confidence intervals of the proposed method much larger than the baselines'? E.g., on FGSM on grb-aminer, the method has a confidence interval of ±0.69, more than ten times larger than any of the baselines. Does this indicate that the proposed model is less stable?
* Removing GraDiv only has a very small effect on the model performance, and is mostly within the confidence interval of the full model. This raises doubts whether GraDiv actually contributes to the performance.

**Novelty**: The PGD-based adversarial training is not new ([Xu et al. 2018], [Geisler et al. 2021]), and the authors do not reference these works. The Mixture of Experts model seems novel in this context, but given my concerns above I am not convinced that it substantially increases the performance.

**Reproducibility**: Limited, given the many open questions that remain for me about the experimental procedure (see above).

**Strength And Weaknesses:**

Strengths:
* Effective adversarial training methods for graph neural networks are rare, so this is an important research area.
* The authors compare to a large number of baselines.
* The authors report strong experimental results.

Weaknesses:
* The authors do not compare to important defense baselines (Soft-Medoid/ Soft-Median [Geisler et al. 2020/2021]) and attacks (PGD attacks [Xu et al. 2018])
* On some datasets there are only n=2 "experts", raising doubts about the usefulness of this contribution.
* The experiment setup is not sufficiently clear (see next section).

**Summary Of The Paper:**

The authors propose a method for robust graph neural networks. The proposed model contains a modified message-passing layer, PGD-based adversarial training, and an additional loss term encouraging diverse node representations. The authors compare their method on several datasets using the GRB benchmark suite.


**Summary Of The Review:**

I have concerns (i) regarding the significance of the individual contributions of the method, (ii) about the lack of important baselines, and (iii) about the experimental setup. Unless these are resolved, I would not recommend publication.

---

> ### Author Response · Authors · 2022-11-13
> **Response to Reviewer SpxS (3/3)**
>
> **Q8**: Similarly in Figure 5, both plots look very similar and it is not clear why the proposed method has an advantage over GCN.
>
> **A8**: In Figure 5, GAME shows more different distributions between adversarial and clean samples than the distributions generated by GCN. This demonstrates that GAME can obtain better distinguishable representations than GCN. We provide a clear visualization in Figure 10 and Appendix F of the revised paper. As shown in Figure 10 (a), almost all the red dots are covered by black dots, which indicates their distributions are too similar and not diverse. This demonstrates the poor capability of GCN in creating adversarial samples. However, in Figure 10 (b), we can see many red dots, given the little overlap between red and black dots.
>
> **Q9**: Table 2: What do the confidence intervals correspond to? Is it standard error/ standard deviation? Over how many training runs? How many different random splits? What is the size of the splits?
>
> **A9**: We follow the settings in GRB [4] and have the following definitions: (1) The confidence intervals correspond to standard deviation; (2) Each result is repeated 10-time runs to report; (3) We use the standard splits provided by the original datasets instead of random splits; (4) The size of the splits (Train:Val:Test) is (0.6:0.1:0.3)$\times$ the size of the dataset. We add more detailed descriptions of these experimental settings in Section 5.2 and 5.3 of the revised paper. Changes are highlighted in red.
>
> [4] Graph Robustness Benchmark: Benchmarking the Adversarial Robustness of Graph Machine Learning, NeurIPS 2021.
>
> **Q10**: Why are the confidence intervals for "w.o. attack" almost always exactly zero for the baselines? Why are the confidence intervals of the proposed method much larger than the baselines'? E.g., on FGSM on grb-aminer, the method has a confidence interval of ±0.69, more than ten times larger than any of the baselines. Does this indicate that the proposed model is less stable?
>
> **A10**: The baselines results in Table 2 are copied from the previous paper [5]. However, we also ran the original code provided by the authors, which reproduces the average accuracy close to [5] but generates a larger standard deviation than that in [5]. The results of some baselines generated by the original code are shown in the following Table 5. From Table 5, we can find that their standard deviations are comparable to ours, which somewhat shows that standard deviations of this magnitude are common.
>
> Table 5: Our reproduced results of w.o. attack baselines on grb-cora dataset. We keep training settings the same as the original paper.
> | Model | GIN+LN | APPNP+LN | GIN+AT | GATGuard | GCN+LN | GAME (ours) |
> |:-------:|:----------:|:---------:|:----------:|:---------:|:----------:|:----------:|
> | Accuracy(\%)  | 77.10$\pm$0.59 | 82.30$\pm$0.55 | 75.21$\pm$0.80 | 66.70$\pm$1.05 | 83.55$\pm$0.62 | 83.55$\pm$0.62 | 88.00$\pm$0.73|
>
>
> [5] Graph Robustness Benchmark: Benchmarking the Adversarial Robustness of Graph Machine Learning, NeurIPS 2021.
>
> **Q11**: Removing GraDiv only has a very small effect on the model performance, and is mostly within the confidence interval of the full model. This raises doubts whether GraDiv actually contributes to the performance.
>
> **A11**: We agree that GraDiv might contribute less significantly compared to the GAME layer and DeCoG. However, when removing GraDiv, the model performance over 10 different runs decreases consistently. This demonstrates the effectiveness of GraDiv in stably improving the model performance. In addition, we visualize the obtained representation of GAME without GraDiv in Figure 8 and Appendix D of the revised paper. To illustrate, in Figure 8 (a) GAME without GraDiv, blue dots are mixed with red and orange dots. However, in Figure 8 (b) GAME, the blue dots, the red dots, and the orange dots are more separated than (a), which shows that GAME including GraDiv performs a very well-clustered embedding space with a more clear inter-cluster difference than GAME without GraDiv, demonstrating the effectiveness of GraDiv in learning distinguishable node representations.

---

> ### Author Response · Authors · 2022-11-13
> **Response to Reviewer SpxS (2/3)**
>
> **Q3**: GraDiv: the additional loss term has complexity $O(N^2)$. This is clearly infeasible for larger datasets, e.g., on the grb-aminer dataset this requires roughly 1.6TB of memory when performing full-batch training. There is no mentioning of batched training, so it is unclear how this additional loss term is used.
>
> **A3**: Thank you for asking this clarification question. We use mini-batch training via neighborhood sampling to train GAME on grb-reddit and grb-aminer datasets, which only require 20235MiB and 23149MiB memory when batch size equals 1024, respectively. We have added more details in Appendix B of the revised paper for better readability. The changes are highlighted in red.
>
>
> **Q4**: DeCoG: What is the difference to standard PGD attacks [Xu et al. 2018]? The fact that only a subset of the GAME model is activated in the forward pass comes from the fact the GAME layer is used, not from the DeCoG contribution itself.
>
> **A4**: The difference between DeCoG and PGD is that PGD setting keeps the number of activated experts unchangeable when generating adversarial samples (Eq. 9 in the paper) and fitting adversarial samples (Eq. 11 in the paper). In particular, in the default PGD setting, only one expert will be activated for the generation and fitting. However, DeCoG leverages and activates all experts for generating adversarial samples, while only activating one expert when fitting the adversarial samples. In addition, we agree that the GAME layer is responsible for the activation of a subset of the GAME model. However, the determination of the number of experts to be activated is obtained from DeCoG. Furthermore, we want to emphasize that DoCoG differentiates the experts between generation and fitting stages.
>
>
> **Q5**: Using infinity-norm is meaningless on graph adjacency matrices because they are discrete, i.e., either we allow no edge to change (epsilon=0) or all edges can change at the same time (epsilon=1).
>
> **A5**: Thanks for pointing out this problem. This is a typo in the paper. Instead, we use $\ell_{0}$ PGD to obtain the discrete edge perturbations, not $\ell_{\infty}$. We have corrected the notation in the revised paper. The changes are highlighted in red.
>
> **Q6**: Figures 6 and 7 are purely qualitative and do not strongly show an advantage of the proposed method over GCN.
>
> **A6**: The quantitative results (i.e., robust accuracy and standard accuracy) comparing GAME and GCN are shown in Section 5.3 and Table 1 of the paper.
> For Figure 6, node representations generated by GCN are generally entangled and intertwined with each other, while GAME can obtain distinguishable node presentations. For example, in Figure 6 (a) GCN, blue dots are mixed with red and orange dots. However, in Figure 6 (b) GAME, some blue dots, red dots, and orange dots are more separated, which shows that GAME exhibits a very well-clustered representation space with a clear inter-cluster difference.
> For Figure 7, our GAME contributes a flatter minima optimization landscape than GCN, which makes the optimization on the GAME model much easier. For instance, in Figure 7 GCN (adversarial), the highest point of the landscape is 20. But in Figure 7 GAME (adversarial) the highest point of the landscape is 15, which is significantly flatter than GCN.
> In addition, visualizing representation space and loss landscape provide insightful investigation for the model properties, and have been widely utilized in previous works [2, 3, 4, 5].
>
> [2] Semi-supervised classification with graph convolutional networks, ICLR 2017
>
> [3] Human-level control through deep reinforcement learning, Nature 2015
>
> [4] Adversarial training for free!, NeurIPS 2019
>
> [5] Decoupled weight decay regularization, ICLR 2019
>
>
> **Q7**: Figure 4 is confusing. In the top row (GCN) the adversarial and clean representations are much more similar than in the bottom row, contrary to the description in the text. This figure shows an advantage of GCN over the proposed method.
>
> **A7**: Thanks for pointing out the concern. We have provided more clarifications and details in Section 5.4 of the revised paper. The changes are highlighted in red. As demonstrated in Figure 1, attacked graphs and clean graphs should have different representation distributions. Therefore, we expect the model can distinguish the mixture distributions of clean and attacked graphs. However, in the top row of Figure 4, we can find that GCN cannot distinguish the difference, where the distribution of clean and attacked graphs are close to each other. On the other hand, in the bottom row of Figure 4, we observe that GAME can clearly separate the distribution of clean and attacked graphs. This demonstrates GAME's outstanding effectiveness against mixture distributions in graph adversarial learning.

---

> ### Author Response · Authors · 2022-11-13
> **Response to Reviewer SpxS (1/3)**
>
> Thank you for the constructive comments. We improve our submission based on your insightful suggestions and provide the following clarifications to your questions. If our answers can address your concerns, we sincerely hope your can increase your score.
>
> **Q1**: The authors do not compare to important defense baselines (Soft-Medoid/Soft-Median [Geisler et al. 2020/2021]) and attacks (PGD attacks [Xu et al. 2018]).
>
> **A1**: Thank you for pointing out the related works. We would like to emphasize that:
> (1) These baselines have different settings to our paper, where our setting following GRB focuses on the black-box scenario, but the baselines [Geisler et al. 2020/2021] focus on the white-box attack. Therefore, their methods cannot be easily compared with our GAME in this manuscript.
> (2) The major contribution of this paper is developing a novel plug-and-play framework that can be easily applied to improve different GNN-based robust learning methods but not proposing a new GNN model that performs better than all existing methods.
> To demonstrate the applicability and effectiveness of our GAME, we introduce the proposed GAME framework into Soft-Medoid/Soft-Median GDCs and show the results in the following Table 1. Also, we introduce the proposed GAME framework into the Robust model [Xu et al. 2018] under PGD attacks and show the results in the following Table 2.
> We can see that our plug-and-play GAME framework is able to improve these methods' performances.
>
> Table 1: The robust performance of Soft Median GDC and Soft Medoid GDC without or with our GAME framework on Cora dataset with $\epsilon=0.1$. We set the number of experts to 10 and the hidden units of each expert to 32.
> | Attack | GR-BCD | PR-BCD |
> |:-------:|:----------:|:---------:|
> | Soft Median GDC [Geisler et al. 2020] | 0.765 ± 0.001 | 0.752 ± 0.002 |
> | Soft Medoid GDC [Geisler et al. 2021] | 0.775 ± 0.003 | 0.761 ± 0.003 |
> | Soft Median GDC (w. GAME) | **0.772 ± 0.005** | **0.759** ± 0.005** |
> | Soft Medoid GDC (w. GAME) | **0.780 ± 0.007** | **0.772** ± 0.006** |
>
> Table 2: Misclassification rates (\%) of robust model [Xu et al. 2018] with or without our GAME framework under PGD attacks. We set the number of experts to 10 and the hidden units of each expert to 32.
> | Dataset | Cora | Citeseer |
> |:-------:|:----------:|:---------:|
> | Robust model [Xu et al. 2018] | 22.0 ± 0.2 | 32.2 ± 0.4 |
> | Robust model (w. GAME) | **21.3 ± 0.4** | **31.5 ± 0.3** |
>
>
> **Q2**: On some datasets there are only n=2 "experts", raising doubts about the usefulness of this contribution. When only one expert is sampled, it could happen that only one of the two experts gets sampled all the time, reverting the model back to a vanilla one.
>
> **A2**: In the common attack setting [1], since the attack is mild, two experts are often sufficient for GAME to defend against the attack samples. We follow Graph Robust Benchmark [1] and adopt this setting in our paper.
> However, in a stronger attack setting, two experts are not enough, and increasing the number of experts improves the performance.
> To illustrate,
> we first conduct experiments on grb-cora with regard to the number of experts in the common attack setting (i.e., the number of nodes and edges attacked is 60 and 20, respectively) and report the results in Table 3.
> From Table 3, we can observe that the performance tends to stay the same when we increase the number of experts in the GAME layer. This shows that two experts are sufficient for GAME to defend against the attack samples.
> Therefore, we set the number of experts to 2 in our paper to avoid intricacy.
>
> Table 3: Results of GAME with different numbers of experts on grb-cora dataset under FGSM attack.
> | num. of experts | 2 | 5 | 10 | 75 | 100 | 250 | 500 | 1000 |
> |:-------:|:----------:|:---------:|:----------:|:---------:|:----------:|:---------:|:----------:|:----------:|
> | GAME            | 0.8653 | 0.8513 | 0.8530 | 0.8610 | 0.8620 | 0.8565 | 0.8535 | 0.8660 |
>
>
> In addition, we show the performance of GAME with respect to the number of experts in a stronger attack setting (i.e., doubling the numbers of attacked nodes and edges to 120 and 40, respectively) in Table 4.
> In particular,
> we observe that
> the performance becomes better when applying more experts.
> This reflects the fact that our GAME is flexible and applicable to different attack intensities.
>
> Table 4: Results of GAME with different numbers of experts on grb-cora dataset under FGSM attack.
> | num. of experts | 2 | 5 | 10 | 75 | 100 | 250 | 500 | 1000 |
> |:-------:|:----------:|:---------:|:----------:|:---------:|:----------:|:---------:|:----------:|:----------:|
> | GAME            | 0.8246 | 0.8234 | 0.8358 | 0.8495 | 0.8470 | 0.8483 | 0.8532 | 0.8545 |
>
>
> At last, for preventing the model from reverting to a vanilla GNN, we employ the noisy gate to avoid the excessive imbalance of expert sampling.
>
> [1] Graph Robustness Benchmark: Benchmarking the Adversarial Robustness of Graph Machine Learning, NeurIPS 2021.

---

> > ### Comment · Reviewer_SpxS · 2022-11-25
> > **Response**
> >
> > Thanks for your reply to my comments. I have a couple of follow-up questions:
> >
> > > These baselines have different settings to our paper, where our setting following GRB focuses on the black-box scenario, but the baselines [Geisler et al. 2020/2021] focus on the white-box attack.
> >
> > Can you explain how the proposed setting is black-box? Are you referring to the attacks or defenses? While the methods GCN as a surrogate model to find attacks, this does not make them black-box: they have access to all data and all labels, i.e., have significant knowledge about the task.
> >
> > What is the reason for not including these methods in Table 2/ Figure 3 of the updated paper, or even referencing them? As of now, my concern is thus not resolved.
> >
> > > The major contribution of this paper is developing a novel plug-and-play framework that can be easily applied to improve different GNN-based robust learning methods but not proposing a new GNN model that performs better than all existing methods.
> >
> > If this is the case, why is GAME compared to all the other GNNs, and uses only a single backbone architecture (GCN)? If the point was to propose a plug-and-play framework, there should be evidence that using it with all kinds of backbone architectures improves results.
> >
> > My views remain unchanged regarding Figures 5, 6, and 7. In my view they do not provide clear insight into the quality of the model.
> >
> > > We use the standard splits provided by the original datasets instead of random splits; (4) The size of the splits (Train:Val:Test) is (0.6:0.1:0.3) x the size of the dataset.
> >
> > Using only a single split to evaluate GNN performance is bad practice and can lead to misleading results, see (noting that a benchmark or previous works do the same does not make it good practice):
> >
> > Shchur, Oleksandr, Maximilian Mumme, Aleksandar Bojchevski, and Stephan Günnemann. "Pitfalls of graph neural network evaluation." Relational Representation Learning Workshop, NeurIPS 2018.
> >
> > Right now, my biggest remaining concerns are (i) not comparing to (or even referencing) two recent robust GNN baselines in the paper, even after my initial review, and (ii) figures contributing little to no evidence about the model quality.

---

> > > ### Author Response · Authors · 2022-11-30
> > > **Further response to Reviewer SpxS (2/2)**
> > >
> > > Q3: My views remain unchanged regarding Figures 5, 6, and 7. In my view they do not provide clear insight into the quality of the model.
> > >
> > > A3: We emphasize that the techniques of visualizing representation distribution, embedding, and loss landscape are pretty popular in prior important works [1, 2, 3, 4, 5, 6]. In particular,
> > > for figure 5, our GAME proves that it creates adversarial samples with more diverse distributions given the same number of iterations (
> > > similar demonstrations of using representation distribution to show the property of different data can be found in previous works [1, 6]);
> > > For figure 6, node representations generated by GCN are generally entangled and intertwined with each other, while GAME can obtain distinguishable node representations (
> > > similar demonstrations of using embedding visualization to evaluate the model quality can be found in previous works [2, 3]);
> > > For figure 7, our GAME's smooth loss landscape demonstrates that the GAME framework can enable GNNs to optimize easily (similar demonstrations of using loss landscape to compare the difficulty of model training are appeared in [4, 5]).
> > >
> > > [1] AugMax: Adversarial Composition of Random Augmentations for Robust Training, NeurIPS 2021
> > >
> > > [2] Semi-supervised classification with graph convolutional networks, ICLR 2017
> > >
> > > [3] Human-level control through deep reinforcement learning, Nature 2015
> > >
> > > [4] Adversarial training for free!, NeurIPS 2019
> > >
> > > [5] Decoupled weight decay regularization, ICLR 2019
> > >
> > > [6] Removing Batch Normalization Boosts Adversarial Training, ICML 2022
> > >
> > > Q4: Using only a single split to evaluate GNN performance is bad practice and can lead to misleading results, see (noting that a benchmark or previous works do the same does not make it good practice):
> > >
> > > A4: In our response in A1 (i.e., Table A, which is comparisons with baselines [Geisler et al. 2020/2021]), we randomly split the train/val/test set and conduct experiments for multiple runs.
> > > In particular,
> > > we first run the GAME framework in three random splits for 3-sigma error of the mean results as shown in [Geisler et al. 2020/2021], which is concluded in Table A. Then, in Table C, we run the experiments 5 times based on different splits of training/testing nodes and report the mean and standard deviation of the misclassification rate like [Xu et al. 2018]).
> > >
> > > Table C: Misclassification rates (\%) of robust model [Xu et al. 2018] with or without our GAME framework under PGD attacks. We set the number of experts to 10 and the hidden units of each expert to 32. We run the experiments 5 times based on different splits of training/testing nodes.
> > >
> > > | Dataset | Cora | Citeseer |
> > > |:-------:|:----------:|:---------:|
> > > | Robust model [Xu et al. 2018] | 22.0 ± 0.2 | 32.2 ± 0.4 |
> > > | Robust model (w. GAME) | **21.3 ± 0.4** | **31.5 ± 0.3** |
> > >
> > > We admit that this paper [7] discovers that the biased results can be caused by a single split on graph datasets which certainly alerts the community. We will include these multi-split results in our final paper.
> > > Also, we believe that the awareness of training with multiple random splits for more accurate comparison needs every member of the graph learning community to emphasize this point.
> > > Thus, we will specifically append a section in the final paper to discuss the random multiple split settings and call for more work to consider and evaluate under this setting.
> > >
> > > [7] Pitfalls of graph neural network evaluation, Relational Representation Learning Workshop, NeurIPS 2018

---

> > > ### Author Response · Authors · 2022-11-30
> > > **Further response to Reviewer SpxS (1/2)**
> > >
> > > Thank you for your additional comments. We improve our final version based on your suggestions about comparing our GAME with other important works (e.g., [Geisler et al. 2020/2021] in white-box settings). More detailed responses to your concerns and questions are as below. If our answer can satisfy your concerns, we truly hope you could consider increasing your score on our paper.
> > >
> > > Q1: How the proposed setting is black-box?
> > >
> > > A1: We would like to clarify that we follow the **definition of black-box setting on attacks** in GRB [1] paper, i.e., “For attackers, they have knowledge about the entire graph (including all nodes, edges, and labels but excluding the labels of the test nodes), but do not have knowledge about the target model or defense mechanism.” and “Attackers are allowed to inject new nodes with limited edges. They are not allowed to modify the original graph for training.”
> > >
> > > However, we must acknowledge that the baselines [Geisler et al. 2020/2021] also provide great insights and fundamental contributions to white-box settings. Thus, we truly appreciate your suggestions, conduct relevant experiments, and will add the results to our paper (since we are not allowed to update these new comparison results in the Openreview system at the current period).
> > > Specifically,
> > > **although we have referenced these works in our current submission (e.g., “Attackers can downgrade the performance of GNNs from multiple perspectives, such as adding or removing edges  [Geisler et al. 2021]”),
> > > we will continue to add the following changes to our paper
> > > **: (i) adding the experimental comparisons with these baselines [Geisler et al. 2020/2021] in Table A into the experiment section,
> > > (ii) highlighting related discussions with these methods [Geisler et al. 2020/2021], i.e., "We introduce the proposed GAME framework into Soft-Medoid/Soft-Median GDCs. We can see that our plug-and-play GAME framework is able to improve these methods' performances."
> > > and (iii) referencing and clarifying more details about white-box settings in the appendix, i.e., "Note that the white-box settings [Geisler et al. 2020/2021] are also equally important to examine GNN's robustness. Thus we provide experimental results in Table A which are repeated 3-time random split for careful comparisons."
> > >
> > > Table A: The robust performance of Soft Median GDC and Soft Medoid GDC without or with our GAME framework on the Cora dataset with the global attacks ($\epsilon=0.1$) proposed in [Geisler et al. 2021]. We set the number of experts to 10 and the hidden units of each expert to 32. We run the GAME framework in three random splits for 3-sigma error of the mean results.
> > >
> > > | Attack | GR-BCD | PR-BCD |
> > > |:-------:|:----------:|:---------:|
> > > |Vanilla GCN| 0.622 ± 0.003 | 0.645 ± 0.002 |
> > > |Vanilla GDC| 0.677 ± 0.005 | 0.674 ± 0.004 |
> > > |Vanilla GCPPRGo| 0.726 ± 0.002 | 0.700 ± 0.002
> > > |Soft Medoid GDC|0.775 ± 0.003 | 0.761 ± 0.003|
> > > |SVD GCN|0.755 ± 0.006 | 0.724 ± 0.006|
> > > |Jaccard GCN|0.664 ± 0.001 | 0.667 ± 0.003|
> > > |RGCN| 0.665 ± 0.005 | 0.664 ± 0.004|
> > > | Soft Median GDC [Geisler et al. 2020] | 0.765 ± 0.001 | 0.752 ± 0.002 |
> > > | Soft Medoid GDC [Geisler et al. 2021] | 0.775 ± 0.003 | 0.761 ± 0.003 |
> > > | Soft Median GDC (w. GAME) | **0.772 ± 0.005** | **0.759 ± 0.005** |
> > > | Soft Medoid GDC (w. GAME) | **0.780 ± 0.007** | **0.772 ± 0.006** |
> > >
> > > [1] Graph Robustness Benchmark: Benchmarking the Adversarial Robustness of Graph Machine Learning, NeurIPS 2021.
> > >
> > > Q2: Why is GAME compared to all the other GNNs, and uses only a single backbone architecture?
> > >
> > > A2: According to Occam's razor, we provide the results of the most widely used and simple GCN architecture in our manuscript, similar to papers [2, 3, 4]. However,
> > > we conduct additional experiments to consider other mainstream GNN architectures such as GAT [5], R-GCN [6], S-GCN [7], GIN [8].
> > > The results are shown in the following Table B.
> > > Table B: Results of the GAME framework with different GNN architectures on grb-cora dataset under FGSM attack.
> > > | GNN arch. | GCN(+AT) | GAT(+AT) | R-GCN(+AT) | S-GCN(+LN) | GIN(+LN) |
> > > |:-------:|:----------:|:---------:|:----------:|:---------:|:----------:|
> > > | Acc. (vanilla)                                | 0.8310 | 0.8533 | 0.8510 | 0.8153 | 0.7826 |
> > > | Acc. (w. GAME 5 experts)           | 0.8513 | 0.8590 | 0.8602 | 0.8266 | 0.8043 |
> > > | Acc. (w. GAME 10 experts)         | 0.8530 | 0.8608 | 0.8585 | 0.8307 | 0.8132 |
> > >
> > > [2] A Unified Lottery Ticket Hypothesis for Graph Neural Networks, ICML 2021
> > >
> > > [3] L2-GCN: Layer-Wise and Learned Efficient Training of Graph Convolutional Networks, CVPR 2020
> > >
> > > [4] Old can be Gold: Better Gradient Flow can make Vanilla-GCNs Great Again, NeurIPS 2022
> > >
> > > [5] Graph Attention Networks, ICLR 2018
> > >
> > > [6] Robust graph convolutional networks against adversarial attacks, KDD 2019
> > >
> > > [7] Simplifying graph convolutional networks, ICML 2019
> > >
> > > [8] How Powerful are Graph Neural Networks? ICLR 2019

---

> > > > ### Comment · Reviewer_SpxS · 2022-11-30
> > > > **Response**
> > > >
> > > > Thanks for your reply.
> > > >
> > > > > However, we must acknowledge that the baselines [Geisler et al. 2020/2021] also provide great insights and fundamental contributions to white-box settings.
> > > >
> > > > They propose defense methods – why does it matter for a **defense** whether it is white- or black-box?
> > > >
> > > > Regarding the black-box definition, this is nowhere to be found neither in the paper nor the supplementary material. How is the reader supposed to know the capabilities of the attacker? This information is so central to the paper that I believe it is not sufficient to refer to the GRB reference only.
> > > >
> > > > > although we have referenced these works in our current submission
> > > >
> > > > This is only true for the 2021 paper, not the 2020 one; and the text only refers to the attack contribution and not the defense.
> > > >
> > > > I appreciate the interesting additional results, thank you for going through the effort. I've increased my score (to "weak reject" due to my remaining concerns), and I am not generally opposed to acceptance given that the other reviewers are much more fond of the submission.

---

> > > > > ### Author Response · Authors · 2022-12-01
> > > > > **Thank you and we will accordingly revise with the plans described below.**
> > > > >
> > > > > Thanks for your constructive suggestions and kind consideration of increasing the evaluation. We will leverage the clarity issue (i.e., a detailed description of the black-box setting) and the reference issue (i.e., an in-depth acknowledgment and discussion of [Geisler et al. 2020/2021] on both the attack and defense strategies) with respect to your comment once we have the chance.
> > > > >
> > > > > The detailed revision plan for our manuscript is described below:
> > > > >
> > > > > **[Clarity Issue]** In section 5.1, under the "Attacking Strategies", we add
> > > > > > These methods are black-box node injection attackers, where information such as gradients, model parameters, and training data is not available. Besides, we also include white-box attackers from Geisler et al. that explore structural perturbation.
> > > > >
> > > > > **[Additional Experiments]** We will merge the additional experiments conducted in this discussion into the final manuscript.
> > > > >
> > > > > **[Additional Discussion for [Geisler et al. 2020/2021]]**
> > > > >
> > > > > * First, we will add proper citations to the **two** papers [Geisler et al. 2020/2021] in the Introduction and Related Work sections for their contributions in both attacking and defensive strategies.
> > > > >
> > > > > * Second, we will extend the discussion in the Related Work section by adding the following supportive descriptions:
> > > > > > Specifically, **[Geisler et al. 2020]** proposed a robust aggregation function motivated by the field of robust statistics in an end-to-end manner, which significantly improves the model's robustness against structural perturbations. Moreover, **[Geisler et al. 2021]** explored sparsity-aware first-order optimization to effectively attack the graph structures and accordingly designed a robust and scalable aggregation function (i.e., Soft Median) to defend GNNs for graphs with different scales.

---

> ### Author Response · Authors · 2022-11-25
> **reminder**
>
> Dear reviewers SpxS,
>
> Thank you again for reviewing our paper. It has been a while since we submitted our responses and we hope your comments have been addressed. If so, we really hope that you can reconsider your evaluation score. Please let us know if you have additional comments. Thank you very much!

---

### Official Review · Reviewer_iMjk · 2022-10-24

**Confidence:** 4
**Correctness:** 4
**Technical Novelty And Significance:** 3
**Empirical Novelty And Significance:** 3
**Recommendation:** 8

**Clarity, Quality, Novelty And Reproducibility:**

The proposed method is well-motivated. Also, the writing is clear and good. The authors have provided sufficient information for reproducibility.


**Details Of Ethics Concerns:**

N.A.

**Strength And Weaknesses:**

Strength:

1. The paper has a clear motivation: the authors take a unified investigation for two clean data and attacked in graph adversarial training. Both are shown to shift to different distributions, but their current GNN-based robust methods do not aware. So, the authors become the first to design a GAME model for unification between the two distributions is necessary to gain robustness.

2. The technical approach is solid: The authors first proposed GAME model to achieve diversity and robustness by adversarial training Graph Mixture of Experts, which has never been used in graph adversarial learning. Because MoE generates different adversarial noise by activating different sub-networks to cooperate with PGD, the GAME model can obtain richer adversarial samples in the same training cycle. Considering that MoE makes the number of parameters larger, to avoid redundancy of features, the authors also designed the regularization module to take full advantage of the different outputs of each expert.

3. Experimental results are SOTA: The GAME model yields significantly improved adversarial robustness, outperforming strong previous methods among small, middle, and large graph datasets. Ablation experiments and analytical experiments are convincing: The authors compared MoE, DECOG, and GRADIV regularization's improvements on robust node classification tasks. The results confirm that MoE, DECOG, and GRADIV regularization are three complementary dimensions, and their proper unification strategy is superior to other current alternatives. The authors also use analytical experiments to reveal that mixed distributions are the important reason for the difficulty of graph adversarial training and that the proposed GAME model handles such mixed distributions well.

4. The paper is well-written and easy to follow. The setup section, appendix section, and anonymous codes provide sufficient details for the implementation of this work.


Weakness:

1. I am not fully convinced why GAME model's computational cost can approximate the normal GNN model.  There was no computational complexity analysis or complexity comparisons with baseline models. Please address this concern.

2. Figure 1 illustrates the GNN trained with clean graphs and the GNN trained with attacked graphs have different representation distributions. Then in Figure 4, the authors state that GAME model can unify clean graphs and attacked graphs. Is this figure plotted based on GAME models (e.g., one GAME model trained with clean graphs and one GAME model trained with attacked graphs)? It seems difficult to obtain enough diversity and enough hardness simultaneously. Please provide more exact descriptions to clarify and support your claim.

**Summary Of The Paper:**

This paper focuses on the adversarial training of robust graph neural networks. The motivation is attractive to me, namely, the divergence of distribution between clean graphs and attacked graphs. This motivation leads to the GAME model, a set of effective all-round designs: leveraging MoE module to construct a more powerful GNN with better capacity, using adversarial training to produce more different attacked samples in the MoE-based GNN, and adding regularization $R_{GraDiv}$ to diversify prediction for more robustness.


**Summary Of The Review:**

The paper is interesting and well-motivated. The proposed model is novel and the experiments are sufficient. There are some minor concerns that need to be addressed.

---

> ### Author Response · Authors · 2022-11-13
> **Response to Reviewer iMjk**
>
> Thanks for your comments. We make clarification to your concern and questions below:
>
> **Q1**: I am not fully convinced why GAME model's computational cost can approximate the normal GNN model. There was no computational complexity analysis or complexity comparisons with baseline models. Please address this concern.
>
> **A1**: We set the dimension of the hidden layer as $L$ for GNN. For our GAME model, we set $N$ as the total number of experts, $d$ as the dimension of a single expert, and $k$ as the number of activated experts. Then we get $L=N\times d$. The computational complexity of normal GNN is $O(L)$ and the computational complexity of GAME's activated parts is $O(k)$. If we fix the $k$ constant, the computation will not grow crazily like normal GNN while the total dimension of the hidden layer grows. Empirical computation measurements are shown as follows:
>
> Table 3: Training time of GCN and GAME with different dimensions. We set the width of a single expert as 32 and $32\times k$ is equal to the total dimension of our GAME model.
> | model   | GCN (32) | GCN (512) | GCN (1024) | GCN (4096) | GAME (32) | GAME (32*16=512) | GAME (32*32=1024) | GAME (32*128=4096) |
> |:-------:|:----------:|:---------:|:---------:|-------:|:----------:|:---------:|:---------:|:---------:|
> | training time | 05:19 | 05:40 | 08:59 | 41:54 | 05:38 | 05:40 | 05:59 | 10:19 |
>
> According to Table 3, when the model's dimension increases, our training time cost increases more slowly than GCN.
>
> **Q2**: Figure 1 illustrates the GNN trained with clean graphs and the GNN trained with attacked graphs have different representation distributions. Then in Figure 4, the authors state that GAME model can unify clean graphs and attacked graphs. Is this figure plotted based on GAME models (e.g., one GAME model trained with clean graphs and one GAME model trained with attacked graphs)? It seems difficult to obtain enough diversity and enough hardness simultaneously. Please provide more exact descriptions to clarify and support your claim.
>
> **A2**: In the bottom row of Figure 4, we use only one GAME model and perform only adversarial training without regular training. We plot the adversarial trained GAME model's two different representation distributions with a clean graph and an attacked graph. The setting of Figure 1 is different from Figure 4. In Figure 1, we show that if one GCN is normally trained and one GCN is adversarially trained, then these two different models will reflect different representation distributions when they are fed a clean graph and an attacked graph, respectively. Based on the conclusion in Figure 1, Figure 4 shows that our GAME can learn two different types of graphs well with only one model.

---

### Official Review · Reviewer_QLnJ · 2022-10-25

**Confidence:** 4
**Correctness:** 4
**Technical Novelty And Significance:** 3
**Empirical Novelty And Significance:** 3
**Recommendation:** 8

**Clarity, Quality, Novelty And Reproducibility:**

The paper is well-organized and the methodology is plainly stated. The proposed model is novel. Code and data sources are available for reproducibility.

**Strength And Weaknesses:**


Strengths:
+The motivation for activating different experts during adversarial loss by PGD is interesting and efficient in model defense. Additionally, the authors recognize and tackle one fundamental issue (mixture distribution of standard and attacked graphs) in adversarial graph learning, which prevents the traditional GNNs from performing well on both adversarial and clean samples. The motivation of this paper is strong, especially as the number of attack methods increases.

+The developed GAME framework is novel. There are three components (GAME layer, DECOG training, and GRADIV regularization) to address each identified problem (model’s generality, training efficiency, and output diversity) correspondingly. The introduced components are not simply combined, but instead form a complementary chain that serves as the overall framework.

+The authors provide a good catch on why graph adversarial learning is challenging for GNNs, which is important and insightful. I am convinced by these statements, e.g., figure 1 presents a clear distribution difference between standard graphs and attacked graphs. The authors also draw the figure in the experiment section to examine the generality of GAME with respect to the difference, which is interesting.

+Extensive experiments and ablation studies are organized to evaluate the proposed GAME in this paper. For instance, the authors evaluate the model in both the attacking setting with five different attack methods and the standard setting with original graph inputs. The proposed model is further validated on different GRB datasets and shows significant results. The results are impressive. The proposed model exhibits state-of-the-art robust accuracy across a variety of attack evaluations and dataset sizes. The proposed model demonstrates remarkable results and superior performances to other methods, which suggests that GAME can be utilized in a wide variety of real-world applications with respect to both performance and reliability.


Weaknesses and Questions:
-Some experimental discussions are not clear for those who are unfamiliar with adversarial learning. For example, in the introduction section, “a progressively larger divergence between the two distributions is observed” is stated in Figure 1’s caption. What does this large divergence mean? The authors should provide more information to explain the definition of divergence or make clarification what divergence means in this experiment.

-According to Section 3, the authors state, "In general, DECOG enables the dynamic activation of each expert in GAME and facilitates the computation of more diverse attacked graph adjacency matrix and node features." I'm curious about the GAME model's diversifying effects with the incorporation of MoE. Will the adversarial training with GAME layers contribute more diverse attack samples than standard GNNs? Will the observation in Section 5.5 still be established on a larger dataset?

-Will a bigger number of activated experts produce a more diverse effect? More discussions are required regarding the activated expert number.


**Summary Of The Paper:**

This paper identifies a principal difficulty of training robust GNNs that existing methods ignore: the convoluted mixture distribution between clean and attacked data samples. To tackle the problem, the authors propose GAME, a novel method for learning robust graph representations from three perspectives. Extensive experiments and analyses demonstrate GAME's superiority over state-of-the-art methods.



**Summary Of The Review:**

By proposing a novel framework comprised of multiple components, this paper addresses three significant difficulties in learning the difference between standard and attacked graphs. The technical contributions are substantial, the motivations are supported by rigorous explanations, and the experimental results are sufficient.

---

> ### Author Response · Authors · 2022-11-13
> **Response to Reviewer QLnJ**
>
> Thanks for your valuable comments. We feel grateful that you acknowledge our writing quality, insights from our experiments, and improvement in robust performance. We improve our manuscript based on your suggestions. More detailed responses to your concerns and questions are as below.
>
> **Q1**: Some experimental discussions are not clear for those who are unfamiliar with adversarial learning. For example, in the introduction section, “a progressively larger divergence between the two distributions is observed” is stated in Figure 1’s caption. What does this large divergence mean? The authors should provide more information to explain the definition of divergence or make clarification what divergence means in this experiment.
>
> **A1**: Divergence in Figure 1's caption refers to the distributional disparities (or differences) between the representations of clean and attacked graphs. “a progressively larger divergence between the two distributions is observed” indicates that after attacked graphs and clean graphs are processed by adversarial trained GCN and regular trained GCN, respectively, the representations of the attacked graphs and the representations of the clean graphs become increasingly different as the model depth increases. For better clarity, we also replace the word "discriminative" with the more commonly used "distinguishable" in the full draft.
>
> **Q2**: According to Section 3, the authors state, "In general, DeCoG enables the dynamic activation of each expert in GAME and facilitates the computation of more diverse attacked graph adjacency matrix and node features." I'm curious about the GAME model's diversifying effects with the incorporation of MoE. Will the adversarial training with GAME layers contribute more diverse attack samples than standard GNNs? Will the observation in Section 5.5 still be established on a larger dataset?
>
> **A2**: We clarify that adversarial training with GAME layers does not create more adversarial samples in terms of quantity; however, for the same number of iterations, our method creates adversarial samples with more diverse adversarial distributions. It is observed in Figure 5, Section 5.5. A similar observation also can be found on the larger dataset grb-flickr, we add this extra experiment in Appendix E.
>
> **Q3**: Will a bigger number of activated experts produce a more diverse effect? More discussions are required regarding the activated expert number.
>
> **A3**: Thanks for your insightful questions. We have the following experimental results on grb-cora to discuss the relationship between the effectiveness and the number of activated experts. The conclusion needs to be discussed in two phrases:
>
> (1)when minimizing loss, we need smaller $k$ (i.e., less activated experts) to help the model optimize more stable, so smaller $k$ (max. loss) brings better results as Table 1 shows (this phenomenon also is discussed in prior work [1]);
>
> Table 1: Results of GAME with different numbers of activated experts during minimizing loss (i.e., fitting adversarial samples) on grb-cora. The number of total experts is 75. The number of activated experts for maximizing loss (i.e., creating adversarial samples) is 3.  We set the attack budgets with 120 nodes and 40 edges.
> | k (min. loss) | 1 | 2 | 5 |
> |:-------:|:----------:|:---------:|:---------:|
> | GAME          | 0.8470 | 0.8333 | 0.8241 |
>
> (2) when generating adversarial samples, we need larger $k$ (i.e., more activated experts) to contribute diverse adversarial gradients, so bigger $k$ (max. loss) brings better results as Table 2 shows.
>
> Table 2: Results of GAME with different numbers of activated experts during maximizing loss (i.e., creating adversarial samples) on grb-cora. The number of total experts is 75. The number of activated experts for minimizing loss (i.e., fitting adversarial samples) is 1. We set the attack budgets with 120 nodes and 40 edges.
> | k (max. loss) | 1 | 2 | 3 | 5 |
> |:-------:|:----------:|:---------:|:---------:|:---------:|
> | GAME            | 0.8305 | 0.8464 | 0.8470 | 0.8475 |
>
> [1] Switch Transformers: Scaling to Trillion Parameter Models with Simple and Efficient Sparsity, JMLR'22

---

### Author Response · Authors · 2022-11-13
**Response to All Reviewers**

We gratefully thank all reviewers for their valuable comments and insightful suggestions. We appreciate that all reviewers recognize the novelty of the proposed GAME framework, e.g., "The developed GAME framework is novel" [by reviewer QLnJ], "The paper is interesting and well-motivated. The proposed model is novel" [by reviewer iMjk], "The Mixture of Experts model seems novel in this context" [by reviewer SpxS], and "The model is relatively new and novel to me." [by reviewer jUru]. Most reviewers concurred that our initial research on the central issue (mixture distributions on graphs) of graph adversarial training is smart and intriguing.

Innovations in algorithms and architectures in graph robust training are evolving rapidly, but in-depth research on the core difficulty of graph adversarial learning problems (mixture distributions on graphs) is also essential. Our experiments about motivation (e.g., Figure 1) facilitate bridging the existing research gap between current methods and the core problem in graph adversarial learning. Based on the motivation to manage complex mixture distributions in adversarial graphs, we design the GAME method, an all-round design with easy-to-use codes for the community. Our work is technically and philosophically innovative, spanning the motivation and methodology.

Reviewers raised several concerns about the paper, such as model setting and model efficiency. To address and clarify these concerns, we carefully run new experimental results with different requirements, modify our manuscripts, and provide responses to your questions and comments. Please let us know if you have any more questions. Thank you!

---

> ### Author Response · Authors · 2022-11-25
> **reminder**
>
> Dear reviewers,
>
> Thank you again for reviewing our paper. It has been a while since we submitted our response. We hope we have addressed your comments and your feedback in the discussion period is important to us.

---

### Decision · Program_Chairs · 2023-01-20

**Decision:**

Accept: poster

**Justification For Why Not Higher Score:**

Since there are still some limitations (see above), I do not recommend a higher score.

**Justification For Why Not Lower Score:**

A rejection might be justifiable if the recommendations are not addressed.

**Metareview: Summary, Strengths And Weaknesses:**

The paper tackles the problem of adversarial robustness of graph representation learning. The authors hypothesize that one reason for the non-robustness is the mixing of adversarial vs. "normale" data distributions. Based on this insight, they propose an improved learning/training principle. The paper addresses and overall important problem and contains some novel insights. Overall, the majority of reviewers indicated acceptance. However, some serious limitations should be addressed by the authors in the final version of the paper:

- The performed split is extremely simple (60% labeled data). The authors should repeat their experiments with a more realistic scenario as done in other works (e.g. only 10% labeled data)
- The authors should highlight the attacker capabilities (e.g. node injection) more clearly. This is not obvious in the current version.
- The comparison to Soft-Medoid/Soft-Median should be performed since these approaches exactly address the "shift" in data distribution as hypothesized by the authors.

If the authors adequately address these points, the paper should be accepted.

**Note From Pc:**

if the above contains the word "oral" or "spotlight" please see: "oral" presentation means -> notable-top-5% and "spotlight" means -> notable-top-25%. As stated in our emails, we are disassociating presentation type from AC recommendations